# RADAR: Learning to Route with Asymmetry-aware DistAnce Representations

**Hang Yi[1], Ziwei Huang[1], Yining Ma[2], Zhiguang Cao[1]**
[1]Singapore Management University, [2]Massachusetts Institute of Technology
`hang.yi.2024@phdcs.smu.edu.sg, ziweihuang@smu.edu.sg`
`yiningma@mit.edu, zgcao@smu.edu.sg`

## Abstract

Recent neural solvers have achieved strong performance on vehicle routing problems (VRPs), yet they mainly assume symmetric Euclidean distances, restricting applicability to real-world scenarios. A core challenge is encoding the relational features in asymmetric distance matrices of VRPs. Early attempts directly encoded these matrices but often failed to produce compact embeddings and generalized poorly at scale. In this paper, we propose RADAR, a scalable neural framework that augments existing neural VRP solvers with the ability to handle asymmetric inputs. RADAR addresses asymmetry from both static and dynamic perspectives. It leverages Singular Value Decomposition (SVD) on the asymmetric distance matrix to initialize compact and generalizable embeddings that inherently encode the *static asymmetry* in the inbound and outbound costs of each node. To further model *dynamic asymmetry* in embedding interactions during encoding, it replaces the standard softmax with Sinkhorn normalization that imposes joint row and column distance awareness in attention weights. Extensive experiments on synthetic and real-world benchmarks across various VRPs show that RADAR outperforms strong baselines on both in-distribution and out-of-distribution instances, demonstrating robust generalization and superior performance in solving asymmetric VRPs.

## 1 Introduction

Vehicle Routing Problem (VRP) represents a classical NP-hard problem in combinatorial optimization with widespread applications such as transportation (Toro O et al., 2016). It requires finding optimal routes to serve spatially distributed customers under various operational constraints. Traditional solvers (Helsgaun, 2017; Lawler & Wood, 1966) either suffer from exponential computational complexity or depend heavily on handcrafted heuristics, which often limit scalability on large-scale VRPs. These challenges have sparked growing interest in neural combinatorial optimization (NCO), which leverages deep learning to develop data-driven solvers that approximate VRP solutions efficiently with reasonable optimality gaps (Huang et al., 2025; Luo et al., 2023). Among those models, constructive neural solvers (Kwon et al., 2020; 2021; Drakulic et al., 2023; 2025; Berto et al., 2025) have proven particularly efficient, as they sequentially select nodes based on partial solutions and instance features, i.e., an approach that closely mirrors the natural process of route construction.

Despite this progress, a significant gap remains between research and real-world applicability. In practice, travel costs are shaped by factors such as road topology, traffic directionality, and physical barriers, introducing asymmetries like one-way streets and time-dependent congestion (Cubukcu & Hatcha, 2016). Yet, most neural VRP solvers assume symmetric Euclidean instances (Kool et al., 2019; Kwon et al., 2020; Luo et al., 2023; Gao et al., 2024; Berto et al., 2025) and rely on coordinate inputs, making them not applicable when only pairwise asymmetric distance matrices are available, as is common in real-world systems. This remains a major bottleneck for deploying NCO models in practical, asymmetric routing scenarios (Son et al., 2026).

To address this limitation, a key challenge is effectively encoding the relational structure of asymmetric distance matrices. In this paper, we consider the modeling of asymmetry into two aspects: *static* asymmetry and *dynamic* asymmetry. Static asymmetry refers to directional discrepancies in the input distance matrix. Current neural solvers incorporate this primarily in two places: encoder attention

and initialization. In attention, many methods concatenate dot-product scores with distance signals to inject static asymmetry information (Kwon et al., 2021). At initialization, however, encoding asymmetry is intrinsically more difficult: asymmetric costs are defined at the edge level, while most architectures operate on node-level representations (e.g., (Drakulic et al., 2023; Luo et al., 2023)). Unlike Euclidean settings, where coordinates provide a geometric scaffold that fully recovers the distance structure, asymmetric matrices lack such geometry, making directional patterns harder to learn. Prior attempts to encode static asymmetry at initialization often lose global structure and thus underperform (Zhou et al., 2025; Son et al., 2026). Dynamic asymmetry refers to learned, layer-dependent interactions differences that emerge inside the encoder's attention. At each layer, the interaction score for $i \rightarrow j$, fused with edge signals, need not equal that for $j \rightarrow i$, and these discrepancies evolve dynamically with context and depth. However, current solvers rely on row-wise softmax attention, which integrates information only over the neighborhood of node $i$. Consequently, $A_{i,j}$ reflects $i$'s local context but ignores how $j$ interacts with the rest of the graph, limiting the model's ability to capture global distance information, particularly in asymmetric settings.

We introduce RADAR: Learning to **R**oute with **A**symmetry-aware **D**ist**A**nce **R**epresentations. RADAR tackles both static and dynamic asymmetries through two key components to learn asymmetry-aware embeddings. For static asymmetry, we propose an initialization scheme based on Singular Value Decomposition (SVD) of the cost matrix. By decomposing it into left and right singular vectors, RADAR learns compact node embeddings that encode each node's role as a source and destination, preserving global directionality. For dynamic asymmetry, we replace the softmax function in the attention mechanism with Sinkhorn normalization (Sinkhorn, 1966), which jointly normalizes rows and columns of the attention matrix. This enforces balanced bidirectional flows, enabling the attention score to capture interactions that depend not only on each node's own neighborhood but also on the neighborhood structure of its counterparts.

Our contributions are as follows: (1) We investigate neural VRP solvers under realistic asymmetric distance matrices, advancing the applicability of NCO methods to real-world scenarios; (2) We introduce an SVD-based initialization that captures global directional relationships from the input distance matrix, improving generalization across instance sizes. We also show that Sinkhorn normalization yields substantial gains, underscoring the importance of capturing the full neighborhood context of both interacting nodes during attention. (3) We evaluate RADAR on 17 synthetic and 3 real-world VRP variants. Across all cases, RADAR consistently outperforms state-of-the-art baselines, demonstrating strong performance. (4) We provide several in-depth analyses, including the structural properties of the input, the role of coordinates under asymmetry, and the evaluation of initialization strategies across different asymmetry levels.

## 2 RELATED WORKS

**Neural Solvers for VRPs.** They generally fall into three categories. *Constructive methods* generate routes sequentially by selecting one node at a time, such as AM (Kool et al., 2019) and POMO (Kwon et al., 2020). *Improvement methods* (Wu et al., 2021; Ma et al., 2021; 2023) begin with an initial solution and learn to iteratively refine it, often combined with classical techniques like local search (Hudson et al., 2022), beam search (Choo et al., 2022), or dynamic programming (Kool et al., 2022). *Heatmap-based methods* (Joshi et al., 2021; Sun & Yang, 2023; Min et al., 2023; Zhang & Cao, 2025) predict pairwise connection probabilities via GNNs and extract solutions from these learned heatmaps using heuristics or post-processing. Among these, constructive approaches are most common but are typically designed for Euclidean VRPs and perform poorly in asymmetric settings.

**Solvers Using Distance Matrices.** Several recent methods (Meng et al., 2025) address the limitations of coordinate-centric neural solvers by directly encoding the distance matrix. MatNet (Kwon et al., 2021) reformulates ATSP (asymmetric traveling salesperson problem) as a bipartite graph, but its one-hot embedding limits scalability. UniCO (Pan et al., 2025) extends this with pseudo one-hot encodings for larger instances. ICAM (Zhou et al., 2025) replaces fixed embeddings with k-nearest distances, enabling generalization to instances with up to 1000 nodes. ReLD (Huang et al., 2025) improves MatNet's decoder with identity mapping and feed-forward layers. RRNCO (Son et al., 2026) incorporates context-aware gating, adaptive biases, and distance-based probabilistic sampling, and further validates its effectiveness on real-world datasets. Post-improvement methods like GLOP (Ye et al., 2024) and UDC (Zheng et al., 2024) adopt divide-and-conquer strategies to

improve scalability. Despite these advances, most models still fail to effectively capture both static and dynamic asymmetry, limiting performance on real-world asymmetric VRPs.

**Graph Positional Encoding.** To embed structural information into node representations, various graph positional encoding techniques have been developed. These methods fall into three main categories: (1) Matrix factorization approaches (Yang et al., 2017; Song et al., 2009; Katz, 1953) compute a proximity matrix (e.g., using spectral or diffusion metrics) and apply dimensionality reduction to derive node embeddings; (2) Random-walk-based methods (Perozzi et al., 2014; Grover & Leskovec, 2016) simulate walks to capture local structure, assigning similar embeddings to frequently co-occurring nodes; (3) Autoencoder-based methods (Wang et al., 2016; Cao et al., 2016) learn low-dimensional embeddings by reconstructing graph structure through neural networks. These techniques inspire embedding strategies in VRP solvers where structural signals, especially under asymmetric cost matrices, need to be effectively encoded.

## 3 PRELIMINARIES

A typical VRP instance comprises a set of nodes, including one or more depots and multiple customers, and is defined by either a distance matrix or coordinates. The objective is to construct a set of routes that (1) visit each customer exactly once, (2) start and end at a depot, (3) satisfy all task-specific constraints (e.g., vehicle capacity), and (4) minimize total travel cost such as the distance. VRP encompasses a wide range of variants depending on the constraints, such as the Capacitated VRP (CVRP), VRP with Time Windows (VRPTW), and Asymmetric VRPs (e.g., ATSP). These variants reflect different practical requirements and significantly increase problem complexity. In this work, we study 17 asymmetric VRP variants. Among them, one is ATSP, and the remaining 16, whose symmetric versions are adapted from RouteFinder (Berto et al., 2025), and we replaced coordinates with asymmetric distance matrices to them. Full details are provided in Appendix A.

## 4 METHODOLOGY

Figure 1 depicts the overall framework of RADAR. Given an asymmetric distance matrix $D$, we compute a truncated SVD $D \approx U_k \Sigma_k V_k^\top$. The left and right factors encode outgoing and incoming signals. We compute a truncated SVD and retain the top-$k$ singular values with their left/right factors $(U_k, \Sigma_k, V_k)$. We form the distance feature by concatenating the left and right components $(U_k \Sigma_k^{1/2}$ and $V_k \Sigma_k^{1/2})$, then concatenate this with node features (e.g., demand) and project the result into the embedding space. We use 5 encoder layers; each has a multi-head attention, two add and normalization, and a feed-forward blocks. In the attention block, $D$ and $D^\top$ are concatenated with the dot product scores, then passed through two linear layers, and normalized by Sinkhorn to obtain doubly stochastic attention scores. At each decoding step, we mask visited nodes and pick the next node by sampling or greedily from the predicted probabilities, until the tour is completed.

### 4.1 STATIC ASYMMETRY: SVD-BASED EMBEDDINGS

A core principle of a successful neural architecture for VRPs is its ability to generate node embeddings that encapsulate both node-specific attributes (e.g., demands) and the relational features (e.g., distance matrix) of the graph. However, we emphasize a crucial point: *most of such designs cannot effectively exploit relational structure unless the initial node embeddings are distinguishable and capture directional distance information*. When all nodes start with identical embeddings, attention outputs remain identical regardless of attention weights, as they are convex combinations over identical value vectors. This renders the model unable to learn effective representations.

Unlike Euclidean problems, where the coordinates of nodes provide strong inductive priors for such node embedding initialization, asymmetric distance matrices lack such geometric structure, making it difficult to generate initial node embeddings to capture their inherent directional patterns. In the absence of informative node features, alternative initialization strategies are required to introduce node-specific variation before relational reasoning begins. Two main approaches have emerged in the literature to address this challenge: uninformed initialization and informed initialization.

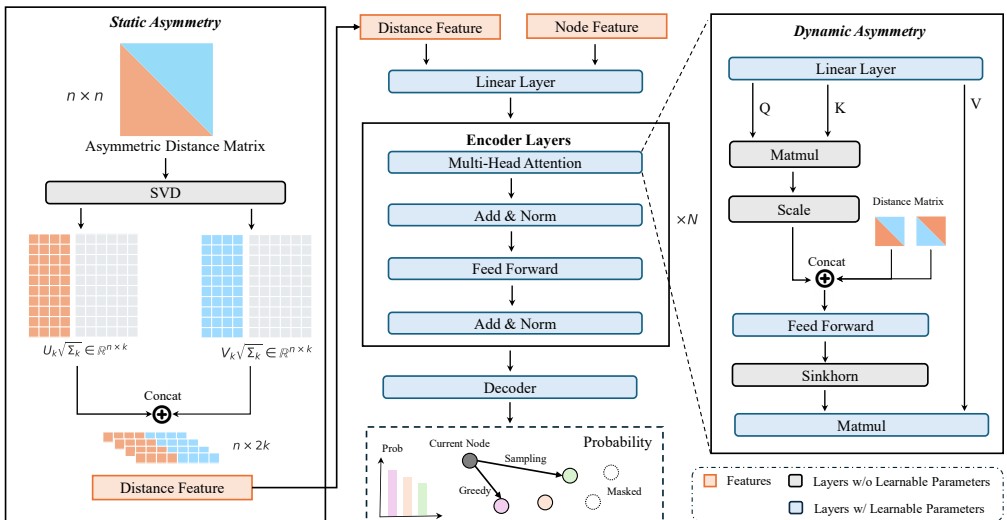

Figure 1: Framework of RADAR, which features two key designs for asymmetric VRPs: (1) an SVD-based embedding method that captures *static asymmetry* in the distance matrix; and (2) Sinkhorn-normalized attention in the encoder layers, replacing standard Softmax to enforce balanced incoming and outgoing attention, modeling *dynamic asymmetry* in representation learning.

**Uninformed initialization** introduces node-distinguishing signals without relying on input structure. These methods provide synthetically distinguishable embeddings, e.g., node indices, fixed positional patterns, or randomly initialized vectors. For example, MatNet (Kwon et al., 2021) initializes row and column embeddings with zero and one-hot vectors. As each node requires a unique one-hot vector in a $d$-dimensional space, the number of nodes $n$ must not exceed $d$. UNICO (Pan et al., 2025) introduces pseudo-hot encoding (POE) to relax the dimensionality constraint for large-scale problems. Although uninformed embeddings can assign distinguishing embeddings, they introduce random noise that lacks any semantic grounding in the input and may break the node symmetry w.r.t. graph structure. As a result, the model must learn to ignore or override meaningless differences, which can hinder learning efficiency and misalign with the task's structural inductive bias.

**Informed initialization** constructs node embeddings by leveraging structural information embedded in input edge features, i.e., the distance matrix $D \in \mathbb{R}^{n \times n}$. This matrix encodes static asymmetry and captures the overall topology of the instance. Rather than injecting arbitrary signals, informed embeddings aim to derive node representations directly from this relational structure, allowing the network to start from relationally meaningful representations. However, informed embeddings also pose unique challenges. The values in the distance matrix $D$ are mutually dependent and collectively define the instance topology. As a result, any projection from edge space to node space can become inherently tied to the number of nodes $n$. This entanglement often leads to size-specific embeddings that generalize poorly to problem instances with different sizes than those seen during training. A common strategy is to summarize each node's neighborhood, e.g., by selecting top-$k$ nearest neighbors (Zhou et al., 2025) or sampling neighbors based on distance-based probabilities (Son et al., 2026), and project the local context into the embedding space. However, such direct use of raw distances as node features fails to capture the underlying topological structure of the graph.

To address the limitations of existing informed embeddings, we focus on the embedding itself and its capacity to encode the structural information of the distance matrices. The inherent asymmetry of the distance matrix is called *static asymmetry*. Ideally, we hope that the initial embeddings could represent the static asymmetry. So, we introduce a formal definition to describe when an embedding can be said to represent static asymmetric relational information.

**Definition 1 (Asymmetry-Aware Embedding).** An embedding matrix $X \in \mathbb{R}^{n \times k}$ is said to be *asymmetric-aware* w.r.t. a feature matrix $D \in \mathbb{R}^{n \times n}$ if there exist two distinct linear transformations $W_1, W_2 \in \mathbb{R}^{k \times k}$ such that

$$\left\| XW_1(XW_2)^\top - D \right\|_F^2 \approx 0. \tag{1}$$

This definition formalizes the capacity of embeddings to represent static asymmetry in a form compatible with attention mechanisms. Attention computes pairwise interactions as $QK^\top$. We adopt a similar bilinear form $XW_1(XW_2)^\top$, with $W_1 \neq W_2$ to produce a non-symmetric interaction matrix that reflects the asymmetry in the original distance matrix. To construct such embeddings, we introduce truncated SVD (TSVD) to reconstruct each node's relative coordinates:

$$D \approx U_k \Sigma_k V_k^\top, \tag{2}$$

where $U_k \in \mathbb{R}^{n \times k}$ and $V_k \in \mathbb{R}^{n \times k}$ contain the top-$k$ left and right singular vectors of $D$, respectively, and $\Sigma_k \in \mathbb{R}^{k \times k}$ is a diagonal matrix consisting of the top-$k$ singular values.

In the cost matrix, each entry $D_{i,j}$ represents the cost from node $i$ (departure) to node $j$ (arrival). Accordingly, the rows of $D$ correspond to departure nodes, and the columns correspond to arrival nodes. Based on this structure, we can construct two intermediate representations $X_L$ and $X_R$, which captures the row-wise features associated and the column-wise features. The distance feature $X \in \mathbb{R}^{n \times 2k}$ is defined as the concatenation of $X_L$ and $X_R$ along the feature dimension:

$$X_L = U_k \sqrt{\Sigma_k}, \quad X_R = V_k \sqrt{\Sigma_k}, \quad X = [X_L \mid X_R]. \tag{3}$$

The obtained $X$ is *asymmetry-aware* w.r.t. $D$ as there exist two projection matrices:

$$W_1 = [I_k \mid 0]^\top \in \mathbb{R}^{2k \times k}, \quad W_2 = [0 \mid I_k]^\top \in \mathbb{R}^{2k \times k} \tag{4}$$

such that

$$XW_1 = U_k \sqrt{\Sigma_k}, \quad XW_2 = V_k \sqrt{\Sigma_k}, \quad XW_1(XW_2)^\top = U_k \Sigma_k V_k^\top \approx D. \tag{5}$$

Therefore, the model is theoretically capturing static asymmetry through a single embedding matrix. The process is shown in Algorithm 1. We analyze the reconstruction quality of the input distance matrix under different truncation levels. The top 10 singular values could capture around 85% of the matrix information, while 20 and 30 singular values improves the retention to about 93% and 97%, respectively. We choose the top 10 singular values as a trade-off between both in-distribution and out-of-distribution generalization. The effectiveness of this choice is further discussed in Section 6.

## 4.2 DYNAMIC ASYMMETRY: SINKHORN NORMALIZATION

In asymmetric VRPs, directional dependencies should be preserved not only at initialization but also during representation learning. In attention-based encoders, node embeddings are updated dynamically layer by layer through weighted aggregation of information from other nodes. The asymmetry learned during encoder attention is called *dynamic asymmetry*. Existing works (Kwon et al., 2021; Drakulic et al., 2025) typically incorporate the distance matrix directly into the computation of attention scores. Formally, the attention from node $i$ to its neighbors is given by:

$$A_{i,:} = \text{Softmax} \left( [\text{Sim}(X_i, X_j, D_{i,j}, D_{j,i})]_{j=1}^N \right), \tag{6}$$

where $\text{Sim}(\cdot)$ denotes a similarity function measuring the compatibility between node $i$ and node $j$ conditioned on both their embeddings and pairwise distances. Compared to vanilla attention, this formulation explicitly integrates node-to-node relational signals $(D_{i,j}, D_{j,i})$ into the similarity function. The subsequent row-wise $\text{Softmax}$ normalizes these scores across all neighbors of $i$, coupling local distance information with broader relational patterns in its neighborhood.

However, such modeling only makes the attention score $A_{i,j}$ aware of distance information in the neighborhood of node $i$ (i.e., $D_{i,:}$ and $D_{:,i}$), while remaining unaware of the complete neighborhood structure of node $j$, i.e., $D_{j,:}$ and $D_{:,j}$. We hypothesize that this limitation weakens the encoder's ability to capture global directional dependencies, since the interaction between $i$ and $j$ is evaluated without considering how $j$ itself relates to the rest of the graph. Notably, this issue is absent in the 2D Euclidean setting, where the entire distance matrix $D$ can be reconstructed from node coordinates, and thus its information is already embedded in the node representations $X$.

To address this, we propose replacing the row-wise $\text{Softmax}$ with *Sinkhorn normalization* (Sinkhorn, 1966), which iteratively normalizes rows and columns of the attention matrix, see Algorithm 2. This

ensures that each attention score $A_{i,j}$ reflects a more complete characterization of both nodes $i$ and $j$, by incorporating the full set of distance-based relations directly connected to them.

---

**Algorithm 1** SVD-based Initialization

**Require:** Distance matrix $D \in \mathbb{R}^{n \times n}$, rank $k$
**Ensure:** Node embeddings $X_{\text{final}}$
 1: $\mu \leftarrow \text{Mean}(D), \sigma \leftarrow \text{Std}(D)$
 2: $D \leftarrow (D - \mu)/\sigma$
 3: $[U, S, V] \leftarrow \text{SVD\_lowrank}(D, k)$
 4: $Q \leftarrow U \cdot \sqrt{S}$
 5: $K \leftarrow V \cdot \sqrt{S}$
 6: $X \leftarrow [Q \mid K]$
 7: $X_{\text{final}} \leftarrow \text{Linear}(X)$
 8: **return** $X_{\text{final}}$

---

**Algorithm 2** Sinkhorn Normalization

**Require:** Score matrix $S \in \mathbb{R}^{n \times n}$, iterations $T$
**Ensure:** Normalized matrix $P$
 1: $P \leftarrow exp(S)$
 2: **for** $t = 1$ to $T$ **do**
 3: $\quad P \leftarrow P/\sum_{\text{col}}(P)$
 4: $\quad P \leftarrow P/\sum_{\text{row}}(P)$
 5: **end for**
 6: **return** $P$

---

## 5 EXPERIMENT

We compare RADAR against baselines on synthetic single-task instances, a multitask setting covering 16 asymmetric VRP variants, and real-world datasets from RRNCO (Son et al., 2026). We then provide several in-depth analyses, examining the effects of coordinates, varying asymmetry levels, and demand distributions, followed by ablation studies. All experiments are run on a single NVIDIA RTX 3090 GPU (24 GB). More details and results are provided in Appendix B and Appendix C. Our code is available at `https://github.com/yihang0410/RADAR`.

### 5.1 SYNTHETIC ATSP AND ACVRP

**Setup.** We evaluate RADAR on synthetic instances of sizes 100, 200, 500, and 1000. All instances are randomly generated following (Kwon et al., 2021; Luo et al., 2023; Kwon et al., 2020). We train RADAR and all baselines on size 100, and evaluate zero-shot generalization to instances of size 200, 500, and 1000 without finetuning. Training takes 39.31h for ATSP and 54.74h for ACVRP.

**Baselines.** *(1) Traditional solvers.* LKH3 (Helsgaun, 2017) with 100, 1k and 10k search trials (10k omitted for ATSP as performance saturates by 100). HGS (Vidal, 2022) under two different time budgets aligned with LKH-1000 and LKH-10000. *(2) Constructive neural solvers.* We retrain MatNet (Kwon et al., 2021), ICAM (Zhou et al., 2025), ELG (Gao et al., 2024), and ReLD (Huang et al., 2025) under our setup; all are evaluated with z-score normalization. We also report MatNet trained for 2100 epochs under its original setting (no z-score). UNICO (Pan et al., 2025) is evaluated using its released checkpoint. We further include MatNet variants: Single (One-hot) replaces the dual embeddings with one-hot vectors, and Single (Random) uses a $\text{Uniform}(0, 1)$ scalar mapped by a linear projection layer. Since ELG does not natively support asymmetry, we adapt it by replacing its encoder with MatNet using random embeddings and removing Euclidean-specific components in the local policy; we then retrain on synthetic data and apply local policy from epoch 1750. In ACVRP, MatNet (Demand) uses only demand as the initial node feature while retaining the original dual–embedding design; MatNet-Single (Demand)$^+$ is a single–branch version that initializes solely from demand. MatNet-Single (Random) follows the same configuration as in ATSP setup. *(3) Improvement Neural Solvers.* GLOP (Ye et al., 2024) is evaluated on our retrained MatNet with z-score but shows no improvement over the authors' original setting. UDC (Zheng et al., 2024) is also retrained in our setup. For fairness, mixed-size training is disabled for both ICAM and UDC.

**Results.** As shown in Table 1, RADAR consistently outperforms prior learning-based baselines. On ATSP, RADAR achieves the lowest objective values and gaps among neural methods, and the gap remains small even as the problem size increases to 1000. On ACVRP, RADAR achieves the best performance among all learning-based methods, and even surpasses LKH on ACVRP200. In Appendix C.1, we also compare RADAR with Matnet and its variants with augmentation on ATSP.

Table 1: Performance on ATSP and ACVRP.

| Method | Testing (1k instances) ATSP100 | | | Generalization (1k instances) ATSP200 | | | ATSP500 | | | ATSP1000 | | |
|---|---|---|---|---|---|---|---|---|---|---|---|---|
| | Obj. | Gap | Time | Obj. | Gap | Time | Obj. | Gap | Time | Obj. | Gap | Time |
| LKH-100 | 1.5643 | 0.00% | 1.08m | 1.5721 | 0.00% | 2.44m | 1.5763 | 0.00% | 7.32m | 1.5739 | 0.00% | 21.92m |
| LKH-1000 | 1.5643 | * | 5.91m | 1.5721 | * | 16.40m | 1.5763 | * | 44.45m | 1.5739 | * | 1.71h |
| GLOP$^+$ | 1.8848 | 20.49% | 2.75m | 2.0584 | 30.93% | 3.07m | 2.2345 | 41.76% | 3.72m | 2.3429 | 48.86% | 7.29m |
| UDC-$x250^+$ ($\alpha = 50$) | 1.5921 | 1.78% | 1.02h | 1.7577 | 11.81% | 2.04h | 2.3810 | 51.05% | 6.41h | 3.0734 | 95.27% | 19.12h |
| Matnet | 1.6473 | 5.31% | 0.03m | 3.1925 | 103.07% | 0.14m | – | – | – | – | – | – |
| Matnet$^+$ | 1.6161 | 3.32% | 0.03m | 1.9111 | 21.56% | 0.14m | – | – | – | – | – | – |
| Matnet-Single (One-hot)$^+$ | 1.5995 | 2.25% | 0.02m | 1.6727 | 6.40% | 0.12m | – | – | – | – | – | – |
| Matnet-Single (Random)$^+$ | 1.5969 | 2.08% | 0.02m | 1.6543 | 5.23% | 0.13m | 1.8610 | 18.10% | 1.34m | 2.1821 | 38.64% | 11.11m |
| MatPOENet-8x$^*$ | 1.8719 | 19.67% | 0.54m | 2.7033 | 71.95% | 2.33m | 4.1189 | 161.30% | 21.87m | 5.5072 | 249.91% | 2.98h |
| ICAM$^+$ | 1.6580 | 5.99% | 0.01m | 1.8471 | 17.49% | 0.06m | 2.4592 | 56.01% | 0.73m | 2.9069 | 84.69% | 9.80m |
| ELG$^+$ | 1.5982 | 2.17% | 0.06m | 1.6423 | 4.47% | 0.27m | 1.7456 | 10.74% | 2.60m | 1.8441 | 17.17% | 22.31m |
| ReLD$^+$ | 1.5900 | 1.64% | 0.03m | 1.6310 | 3.75% | 0.15m | 1.7873 | 13.39% | 1.49m | 2.0723 | 31.67% | 12.18m |
| RADAR$^+$ | **1.5756** | **0.72%** | 0.04m | **1.5879** | **1.01%** | 0.15m | **1.6098** | **2.13%** | 1.45m | **1.6389** | **4.13%** | 11.57m |

| Method | Testing (1k instances) ACVRP100 | | | Generalization (1k instances) ACVRP200 | | | ACVRP500 | | | ACVRP1000 | | |
|---|---|---|---|---|---|---|---|---|---|---|---|---|
| | Obj. | Gap | Time | Obj. | Gap | Time | Obj. | Gap | Time | Obj. | Gap | Time |
| LKH-100 | 2.2526 | 6.05% | 2.20m | 2.2245 | 2.77% | 3.24m | 2.4155 | 3.20% | 11.58m | 2.5092 | 2.84% | 1.00h |
| LKH-1000 | 2.1635 | 1.86% | 17.64m | 2.1807 | 0.75% | 25.32m | 2.3605 | 0.86% | 1.06h | 2.4569 | 0.69% | 2.86h |
| LKH-10000 | 2.1240 | 0.00% | 2.79h | 2.1645 | 0.00% | 4.25h | 2.3405 | 0.00% | 10.29h | 2.4400 | 0.00% | 36.25h |
| HGS-Short$^\#$ | 2.1614 | 1.76% | 16.67m | 2.0806 | -3.88% | 25.38m | 2.3011 | -1.68% | 1.08h | 2.1094 | -13.55% | 2.99h |
| HGS-Long$^\#$ | 2.0942 | -1.40% | 2.78h | 1.9733 | -8.83% | 4.24h | 2.1451 | -8.35% | 10.47h | 1.9792 | -18.89% | 36.33h |
| Matnet (Demand)$^+$ | 2.1968 | 3.42% | 0.04m | 3.1620 | 46.10% | 0.20m | 4.4847 | 91.58% | 1.92m | 5.7523 | 135.76% | 13.59m |
| Matnet-Single (Demand)$^+$ | 2.1821 | 2.73% | 0.03m | 2.6283 | 21.43% | 0.15m | 3.4994 | 49.46% | 1.67m | 4.3736 | 79.25% | 11.79m |
| Matnet-Single (Random)$^+$ | 2.1813 | 2.70% | 0.03m | 2.2372 | 3.36% | 0.15m | 3.3697 | 43.98% | 1.73m | 3.9904 | 63.52% | 11.38m |
| ReLD$^+$ | 2.1656 | 1.96% | 0.04m | 2.1635 | -0.05% | 0.18m | 3.4507 | 47.40% | 1.96m | 4.7424 | 94.45% | 13.80m |
| RADAR$^+$ | **2.1588** | **1.64%** | 0.05m | **2.1483** | **-0.75%** | 0.14m | **2.4198** | **3.39%** | 1.75m | **2.4634** | **0.96%** | 11.60m |

**Note.** $^*$ denotes evaluation using the authors' official checkpoints; $^+$ indicates training and testing with z-score normalization. $^\#$ indicates that HGS yields infeasible solutions under the given time budgets. Consequently, we do not use it as the baseline for gap computation, and the detailed infeasible rates are reported in Appendix G. Results for UDC and ICAM differ from their original reports because we disabled their mixed-size training scheme to ensure fairness. MatNet with one-hot node embeddings fails to generalize to $N \in \{500, 1000\}$, as the identity embedding is tied to a fixed 256-dimensional space. The boldface indicates the best result among learning-based methods.

## 5.2 MULTI-TASK LEARNING ON VARIOUS ASYMMETRIC VRP VARIANTS

**Setup.** We further integrate our RADAR into the multi-task framework in RouteFinder (RF) (Berto et al., 2025) and follow its experimental setup to demonstrate its strong generalizability. The model is trained on 16 asymmetric VRP variants and evaluated on 1,000 test instances per variant.

**Baselines.** We adopt the traditional solver HGS (Vidal, 2022). For neural baselines, we adapt ROUTEFINDER with two changes: (i) replace its encoder attention with MatNet attention; and (ii) modify the initialization, yielding two variants, i.e., **RF**, which uses no distance features and initializes solely from node attributes (e.g., demand, time windows), and **RF-NN**, which uses top-k nearest-neighbor distances concatenated with the node features.

Table 2: Average performance.

| Method | Avg. Obj. | Avg. Gap (%) |
|---|---|---|
| HGS | 2.4709 | – |
| OR-Tools | 2.5409 | 3.0863 |
| RF | 2.5330 | 2.4688 |
| RF-NN | 2.5216 | 1.9875 |
| RADAR | **2.5047** | **1.3331** |

**Results.** Table 2 summarizes average performance over 16 asymmetric VRP variants in the multitask setting. Despite the diversity of constraints, RADAR remains competitive and achieves the lowest average gap among all neural methods. This validates RADAR's asymmetry-centric design across a broad spectrum of asymmetric routing problems. See Table 8 for more results.

## 5.3 REAL WORLD DATASETS

We evaluate RADAR on real-world asymmetric benchmarks introduced by RRNCO (Son et al., 2026). Following their framework, we train our model on the same datasets using Min-Max normalization. Since the test sets remain unchanged, we directly reuse the GCN and MatNet results reported in their paper, excluding other baselines due to incompatible settings. Unlike synthetic datasets, RRNCO provides both realistic node coordinates and asymmetric distance matrices (Section 5.4 further analyzes their individual impact). Table 3 shows the results across three real-world tasks. RADAR consistently achieves lower costs and smaller optimality gaps across all tasks and distribution settings. It also generalizes well to both in-distribution and out-of-distribution instances, highlighting

Table 3: Performance on Real-world Datasets for ATSP, ACVRP, and ACVRPTW.

| Task | Method | In-distribution | | | Out-of-distribution (city) | | | Out-of-distribution (cluster) | | |
|---|---|---|---|---|---|---|---|---|---|---|
| | | Cost | Gap (%) | Time | Cost | Gap (%) | Time | Cost | Gap (%) | Time |
| ATSP | LKH3 | 38.387 | * | 1.6h | 38.903 | * | 1.6h | 12.170 | * | 1.6h |
| | MatNet | 39.915 | 3.98 | 27s | 40.548 | 4.23 | 27s | 12.886 | 5.88 | 27s |
| | RRNCO | 39.077 | 1.80 | 21s | 39.783 | 2.26 | 21s | 12.450 | 2.30 | 21s |
| | RADAR | **38.671** | **0.74** | 21s | **39.272** | **0.95** | 21s | **12.314** | **1.18** | 21s |
| ACVRP | PyVRP | 69.739 | * | 7h | 70.488 | * | 7h | 22.553 | * | 7h |
| | OR-Tools | 72.597 | 4.10 | 7h | 73.286 | 3.97 | 7h | 23.576 | 4.54 | 7h |
| | GCN | 90.546 | 29.84 | 17s | 90.805 | 28.82 | 17s | 34.417 | 52.61 | 17s |
| | MatNet | 74.801 | 7.26 | 30s | 75.722 | 7.43 | 30s | 24.844 | 10.16 | 30s |
| | RRNCO | 72.145 | 3.45 | 23s | 72.999 | 3.56 | 23s | 23.280 | 3.22 | 23s |
| | RADAR | **71.557** | **2.61** | 23s | **72.336** | **2.62** | 23s | **23.137** | **2.59** | 23s |
| ACVRPTW | PyVRP | 118.056 | * | 7h | 118.513 | * | 7h | 39.253 | * | 7h |
| | OR-Tools | 119.681 | 1.38 | 7h | 120.147 | 1.38 | 7h | 39.903 | 1.66 | 7h |
| | RRNCO | 122.693 | 3.93 | 32s | 123.249 | 4.00 | 32s | 41.077 | 4.65 | 32s |
| | RADAR | **121.260** | **2.71** | 32s | **121.841** | **2.81** | 32s | **40.459** | **3.07** | 32s |

**Note.** The boldface indicates the best result among learning-based methods.

Table 4: Study on the effect of coordinates in asymmetric settings (ATSP100).

| Method | In-distribution | | | Out-of-distribution (city) | | | Out-of-distribution (cluster) | | |
|---|---|---|---|---|---|---|---|---|---|
| | Cost | Gap | Time | Cost | Gap | Time | Cost | Gap | Time |
| LKH | 38.387 | * | 1.6h | 38.903 | * | 1.6h | 12.170 | * | 1.6h |
| RRNCO (w/o coords) | 39.624 | 3.22% | 7s | 40.670 | 4.53% | 7s | 12.982 | 6.68% | 7s |
| RRNCO (w/o coords + aug) | 39.309 | 2.40% | 21s | 40.020 | 2.87% | 21s | 12.567 | 3.26% | 21s |
| RADAR (w/o coords) | 38.958 | 1.49% | 7s | 39.551 | 1.66% | 7s | 12.413 | 2.00% | 7s |
| RRNCO (w/ coords) | 39.385 | 2.60% | 7s | 40.383 | 3.80% | 7s | 12.717 | 4.50% | 7s |
| RRNCO (w/ coords + aug) | 39.077 | 1.80% | 21s | 39.783 | 2.26% | 21s | 12.450 | 2.30% | 21s |
| RADAR (w/ coords) | 38.972 | 1.52% | 7s | 39.765 | 2.22% | 7s | 12.487 | 2.61% | 7s |
| RADAR (w/ coords + aug) | 38.671 | 0.74% | 21s | 39.272 | 0.95% | 21s | 12.314 | 1.18% | 21s |

its robustness under real-world variability. These results underscore RADAR's practicality and effectiveness in handling complex, asymmetric routing in real-world scenarios.

## 5.4 COORDINATES VS. DISTANCE MATRICES

We study whether coordinates provide meaningful signals in asymmetric routing tasks. Following the ATSP setup from RRNCO (Son et al., 2026), we can isolate the effect of coordinates. We compare RADAR and RRNCO under various coordinate usage conditions, applying the augmentation strategy from POMO (Kwon et al., 2020). Since RADAR employs deterministic initialization, the (w/o coords + aug) case yields identical outputs and is therefore omitted. Results show that RRNCO degrades noticeably without coordinates, while RADAR maintains strong performance. Remarkably, even without coordinates, RADAR outperforms RRNCO with coordinate augmentation, indicating that our distance-based embeddings effectively capture structural information without positional cues. When coordinate augmentation is applied, both methods improve, with RADAR being the best. This shows that, in asymmetric tasks, the main value of coordinates may lie in enabling augmentation and promoting diversity, rather than encoding structure.

## 5.5 EFFECT OF INITIALIZATION UNDER DIFFERENT ASYMMETRY LEVELS

In real-world scenarios, distance matrices often violate geometric assumptions such as the triangle inequality. As asymmetry increases, structural regularity deteriorates, making it harder for models to extract meaningful representations. Since initialization influences early learning dynamics, it is important to understand how varying asymmetry levels impact embedding quality. To simulate this, we generate node coordinates uniformly in $[0, 1]^2$ and compute pairwise Euclidean distances

Table 5: Cost and relative GAP (%) under varying asymmetry levels. RADAR serves as the baseline.

| Method | Low Asymmetry | | | | Medium Asymmetry | | | | High Asymmetry | | | |
|---|---|---|---|---|---|---|---|---|---|---|---|---|
| | 50 | GAP | 100 | GAP | 50 | GAP | 100 | GAP | 50 | GAP | 100 | GAP |
| MatNet[†] | 4.6446 | 0.54% | 6.2992 | 1.83% | 4.8124 | 9.55% | 6.7598 | 11.84% | 4.7877 | 21.93% | 6.9756 | 24.04% |
| Random[†] | 4.6851 | 1.41% | 6.3244 | 2.24% | 4.5128 | 2.72% | 6.2680 | 3.70% | 4.0722 | 3.70% | 5.9836 | 6.41% |
| UNICO[†] | 4.6235 | 0.08% | 6.4936 | 4.98% | 4.4896 | 2.19% | 7.2094 | 19.27% | 4.0759 | 3.80% | 6.5876 | 17.14% |
| ICAM[‡] | 4.6966 | 1.66% | 6.3632 | 2.87% | 4.4840 | 2.06% | 6.2293 | 3.05% | 4.0636 | 3.48% | 5.9471 | 5.75% |
| RRNCO[‡] | 4.6995 | 1.73% | 6.4048 | 3.54% | 4.5005 | 2.44% | 6.3436 | 4.95% | 4.1282 | 5.13% | 6.0893 | 8.28% |
| **RADAR[‡]** | **4.6198** | * | **6.1858** | * | **4.3933** | * | **6.0444** | * | **3.9268** | * | **5.6235** | * |

**Note.** [†] indicates *uninformed* embedding; [‡] indicates *informed* embedding.

to form a symmetric matrix. Asymmetry is introduced by multiplying each entry $D_{ij}$ with a noise coefficient $\theta_{ij}$ sampled from $\mathcal{N}(1, \sigma^2)$, where $\sigma$ controls asymmetry: weak ($\sigma = 0.1$), moderate ($\sigma = 0.2$), and strong ($\sigma = 0.3$). We compare initialization strategies from RRNCO (Son et al., 2026), UniCO (Pan et al., 2025), MatNet (Kwon et al., 2021), ICAM (Zhou et al., 2025), and random initialization. For RRNCO and ICAM, we use single-embedding variants without coordinate inputs to isolate initialization effects. All methods are evaluated using a unified MatNet-style attention architecture, trained on 50-node instances and tested on 50- and 100-node sets. Training runs for 1200 epochs, with performance recorded at epoch 1101. Additional settings follow Section 5.1.

The results in Table 5 presents the performance of different initialization strategies under increasing levels of asymmetry. Overall, performance tends to degrade as asymmetry intensifies, but the rate of degradation varies significantly across methods. Approaches such as MatNet and UniCO ([†]) exhibit sharp increases in relative gaps, particularly at larger problem sizes and higher asymmetry levels. In contrast, informed initialization methods ([‡]) degrade more gradually and maintain lower overall gaps. Notably, RADAR demonstrates consistently strong performance across all settings.

### 5.6 DIFFERENT DEMAND DISTRIBUTION

In VRP models, customer demands are typically sampled from a discrete uniform distribution over $\{1, \ldots, 9\}$. While this demand setting is convenient for training and evaluation, it provides only a limited view of the demand landscape. A model that fits well under this single distribution may lack sensitivity to distributional shifts, leading to degraded performance once the demand law deviates from the training regime. See Appendix C.3 Table 9 for more details.

## 6 ABLATION STUDY AND DISCUSSIONS

### 6.1 SVD-BASED INITIALIZATION

**Current Initialization Methods.** We evaluate the proposed initialization against ICAM (Zhou et al., 2025), UniCO (Pan et al., 2025), Random, MatNet (Kwon et al., 2021), and RRNCO (Son et al., 2026) under the same experimental setup described in Section 5.5. UniCO could not run on ATSP1000 due to memory constraints. As shown in Figure 2, the proposed initialization yields consistent gains on in-distribution instances and exhibits stronger out-of-distribution generalization to larger problem sizes, outperforming all baselines in both regimes.

**Alternative SVD Initialization Methods.** We compare our SVD-based initialization with eigenvalue decomposition (EVD), multidimensional scaling (MDS) (Beals et al., 1968), QR decomposition (Francis, 1961), and random approximation (RA) variants on ATSP100–1000. As evidenced by Table 10 and the theoretical analysis in Appendix D.2, our method aligns best with the asymmetric structure of the distance matrices, leading to superior performance over existing alternatives.

**Effect of $k$ in Informed Embedding.** We further examine the effect of $k$, the number of informative neighbors used during initialization. We apply this analysis to ICAM, RRNCO, and our method with $k \in \{10, 30, 50\}$. Figure 3 shows radar plots of generalization performance. We report an *Efficiency Score*, defined as $\max(1 - \text{Gap}, 0)$, to capture both accuracy and stability. Increasing $k$

Table 6: Ablation on SVD and Sinkhorn for RADAR.

| Method | SVD | Sinkhorn | 100 | Gap(%) | Time | 200 | Gap(%) | Time | 500 | Gap(%) | Time | 1000 | Gap(%) | Time |
|--------|-----|----------|-----|--------|------|-----|--------|------|-----|--------|------|------|--------|------|
| RADAR | ✗ | ✗ | 1.5969 | 2.08 | 0.02m | 1.6543 | 5.23 | 0.13m | 1.8610 | 18.06 | 1.34m | 2.1821 | 38.64 | 11.11m |
| RADAR | ✓ | ✗ | 1.5928 | 1.82 | 0.03m | 1.6273 | 3.51 | 0.14m | 1.7418 | 10.50 | 1.44m | 1.9342 | 22.89 | 11.45m |
| RADAR | ✗ | ✓ | 1.5829 | 1.19 | 0.03m | 1.6007 | 1.82 | 0.13m | 1.6379 | 3.91 | 1.43m | 1.6878 | 7.24 | 11.37m |
| RADAR | ✓ | ✓ | **1.5756** | **0.72** | 0.04m | **1.5879** | **1.01** | 0.15m | **1.6098** | **2.13** | 1.45m | **1.6389** | **4.13** | 11.57m |

improves in-distribution accuracy but tends to degrade generalization to larger instances. Since our method remains low-cost and stable across all tested $k$, we fix top-10, which offers the strongest generalization while retaining competitive in-distribution performance.

**Runtime Ananlysis.** We profile the SVD step to characterize its computational cost. Our implementation employs a randomized, truncated SVD that is GPU-accelerated and supports batched evaluation across instances. As shown in Figure. 4 and discussed in Appendix D.4, end-to-end runtime scales smoothly with problem size, and SVD becomes progressively less dominant at larger scales.

## 6.2 SINKHORN NORMALIZATION

**Sinkhorn Normalization.** To isolate the impact of Sinkhorn normalization, we compare RADAR with and without it. As shown in Table 6, incorporating Sinkhorn significantly improves performance across all instance sizes, highlighting its importance in modeling directional flow consistency.

**Sinkhorn VS. Softmax.** To highlight the advantage of Sinkhorn over the standard softmax normalization, we conduct an additional comparison on ATSP100. Specifically, we contrast the first 10 epochs and the last 10 epochs of training, as reported in Appendix D.5. The results show that Sinkhorn enables faster convergence and yields better performance on asymmetric problems.

**Runtime Analysis.** We further profile the Sinkhorn normalization on ATSP100 (see Fig. 4 and Appendix D.6). Our measurements separate end-to-end wall time from module breakdowns. The GPU-batched implementation keeps Sinkhorn's overhead modest and stable across problem sizes.

**Number of Iterations.** In all main experiments we set the number of Sinkhorn iterations to T = 10. In Appendix D.7, we conduct a sensitivity study on ATSP with iterations.

## 7 CONCLUSION

This work introduces a framework for effectively incorporating asymmetric distance information into neural VRP solvers. By leveraging SVD-assisted structured initialization and Sinkhorn-normalized attention, our approach achieves strong and consistent performance across a wide range of asymmetric VRP variants. More importantly, it demonstrates improved generalization and robustness in modeling asymmetric structures, making it well-suited for the decision-making in real-world VRP scenarios. In future work, we aim to steer NCO toward more realistic settings. First, we will extend RADAR to a broader class of combinatorial optimization problems in which *relational (edge) features* are central. Our recipe use a SVD-based initialization to turn edge signals into compact node-level anchors. We expect this to improve cold-start quality. Second, we plan to integrate RADAR with richer paradigms, such as improvement heuristics and search-based solvers, where RADAR provides asymmetric, cost-aware priors to guide neighborhood selection and move acceptance. We believe these directions will further enhance robustness under distribution shift and strengthen the application of NCO models in solving real-wolrd VRPs.

## ACKNOWLEDGMENTS

This research is supported by the National Research Foundation, Singapore under its AI Singapore AI Research Fundamental Research Collaborative (US-NSF Researcher Call) (AISG Award No: AISG3-RP-2025-036-USNSF) and its AI Singapore Programme (AISG Award No: AISG3-RP-2022-031). Any opinions, findings and conclusions or recommendations expressed in this material are those of the author(s) and do not reflect the views of National Research Foundation, Singapore. This research/project was supported by the Singapore Ministry of Education (MOE) Academic Research Fund (AcRF) Tier 1 grant (Grant Approval No: 22-SIS-SMU-064).

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

# A PROBLEM DETAILS

This section defines the five key constraints used in the 17 asymmetric VRP variants. These variants are commonly used in previous work (e.g., (Zhou et al., 2024; Berto et al., 2025)) as benchmarks to evaluate the applicability and multitask performance of learned VRP solvers. We follow the setup in a recent work, RouteFinder (Berto et al., 2025), to generate instances for each variant. To better reflect real-world conditions such as one-way streets and traffic patterns, all problems are made **asymmetric**, meaning the travel cost from node $i$ to node $j$ may differ from the cost from $j$ to $i$, consistent with the MatNet setting (Kwon et al., 2021). Note that the term *ACVRPTW* used in Section 5.3 is equivalent to *AVRPTW* in this table.

**Open route (O).** The vehicle does not need to return to the starting point after visiting all customers.

**Capacity (C).** Under this constraint, each customer is associated with a fixed nonnegative demand, representing the quantity of goods to be delivered. Each vehicle has a predefined capacity limit, which denotes the maximum total demand it can serve on a single route. A route is considered feasible only if the sum of the demands of all customers assigned to it does not exceed the vehicle capacity. This constraint ensures that vehicles do not operate beyond their load limits.

**Backhauls (B).** In the backhaul constraint, customers are divided into two groups: linehaul customers, who need goods delivered from the depot (positive demand), and backhaul customers, who need goods picked up and brought back to the depot (negative demand). The constraint requires that all deliveries to linehaul customers must be completed before starting any pickups from backhaul customers on the same route. This rule limits the order in which customers can be visited.

**Duration limit (L).** Each vehicle is limited by a maximum travel time or distance. The total time or distance of the route must be within this limit. If the route exceeds the limit, it is considered infeasible.

**Time window (TW).** The time window constraint assigns each customer a time interval during which the visit must occur. A vehicle can only serve a customer if it arrives within the time window. If it arrives early, it must wait until the window opens; if it arrives late, the service is considered infeasible.

# B EXPERIMENT DETAILS

## B.1 DETAILS OF ATSP AND ACVRP

**Problem setting.** We construct distance matrices following MatNet (Kwon et al., 2021) and enforce the triangle inequality to ensure metric consistency. We perform Z-score normalization on distance matrices, where each entry is transformed by subtracting the mean and dividing by the standard deviation of pairwise distances. Customer demands are randomly sampled as integers from 1 to 9, based on LEHD (Luo et al., 2023) and POMO (Kwon et al., 2020), and normalized by their total sum. Vehicle capacities are set at 50, 80, 100, and 250 for ACVRP instances of sizes 100, 200, 500, and 1000, respectively, following LEHD (Luo et al., 2023).

**Training Setting.** The embedding dimension is set to 256, with 8 attention heads and 5 encoder layers. We use the Adam optimizer with an initial learning rate of $4 \times 10^{-4}$ and a weight decay of $10^{-6}$. The model is trained for a total of 2100 epochs. A learning rate scheduler is employed, maintaining a constant learning rate before epoch 2001, after which a decay factor of $\gamma = 0.1$ is applied. Each epoch consists of 10,000 instances and a batch size of 64. Compared to MatNet (Kwon et al., 2021), which employs 16 attention heads and 12000 training epochs, fewer attention heads and epochs are used in our RADAR to reduce memory usage and improve training efficiency. In addition, bias terms are introduced in the projection matrices $W_q$ and $W_k$. As a result, our training process achieves a 89% reduction in training time compared to MatNet.

**Testing Setting.** A greedy decoding strategy is used for inference. Following MatNet (Kwon et al., 2021), the number of trajectories is set equal to the number of nodes. Each test set contains 1000 instances, and the reported inference time reflects the total wall-clock time for the full test set. The test batch sizes are configured to fully use the memory of one GPU card as follows: 1000 for ATSP100, 400 for ATSP200, 50 for ATSP500, and 3 for ATSP1000.

**Baseline.** The selected baselines represent both strong traditional and state-of-the-art reinforcement learning-based methods, each reporting performance surpassing that of MatNet (Kwon et al., 2021) to handle asymmetric VRPs. All constructive neural solvers are retrained using our training setup, except UNICO (Pan et al., 2025), where the released checkpoint is used. Since UNICO was trained on the same distribution as our test data, the provided checkpoint remains representative for fair comparison. These baselines reflect the representative design choices of learning-based approaches to handle asymmetric VRPs in the current literature. MatNet and MatPOENet-8x adopt dual embedding architectures, which separately encode source and target nodes, and can naturally capture directional distance information. Therefore, they do not require the transposed distance matrix in encoder attention. Other constructive neural solvers use single embeddings and typically incorporate the transposed distance matrix to model asymmetry.

**Our Model.** We build upon Matnet (Kwon et al., 2021) by replacing its original dual embedding with our SVD-based initialization and substituting the softmax with Sinkhorn normalization in the encoder.

### B.2 DETAILS OF MULTI-TASK VRP

**Problem Setting.** All problem instances are generated following Routefinder (Berto et al., 2025), except for the coordinates, which are replaced by an asymmetric distance matrix constructed based on MatNet (Kwon et al., 2021) and standardized via z-score normalization. However, enforcing strict satisfaction of the triangle inequality across all datasets becomes computationally expensive during instance generation in the training phase. To mitigate this, we relax the constraint by applying a correction: for each node triplet, the direct distance is updated as $D_{ij} = \min(D_{ij}, D_{ik} + D_{kj})$ over $n$ iterations. This approximation is reasonable in the context of hard benchmarks, as real-world routing data often deviates from a strict triangle inequality. Service times are uniformly sampled in the range $[0.12, 0.15]$, and time window lengths are drawn from $[0.15, 0.3]$. Other problem settings follow Routefinder (Berto et al., 2025).

**Training Setting.** We follow the same training setting as Routefinder Berto et al. (2025). We train each model for 300 epochs using dynamically generated VRP instances, with 100,000 samples per epoch. The optimization is performed by the Adam algorithm with an initial learning rate of $3 \times 10^{-4}$ and a batch size of 256. The learning rate is reduced by a factor of 0.1 at epochs 270 and 295.

**Testing Setting.** We use the same evaluation setting as RouteFinder with greedy decoding. For each instance, we generate as many solution trajectories as the number of nodes as (Kwon et al., 2020) and (Kwon et al., 2021). Each variant's test set consists of 1000 instances of size 100, and the reported inference time is measured over the entire set.

**Baseline.** Since the original RouteFinder architecture is not designed for asymmetric vehicle routing problems, we introduce two modified variants, RF and RF-NN, which represent the minimal changes required to adapt it to our setting. Both variants are implemented within the RouteFinder framework and incorporate MatNet attention to model asymmetric relationships. In RF, each node is initialized using only its own node features, such as demand and service time window, without access to any pairwise distance information. RF-NN extends this by incorporating the distances to the 50 nearest neighbors as additional node features, following the initialization used in ICAM (Zhou et al., 2025).

**Our Model.** To verify the effectiveness of our method, we build upon the RouteFinder framework by replacing its original coordinate embedding with our SVD-based initialization and modifying the attention mechanism to Matnet attention with Sinkhorn normalization.

### B.3 DETAILS OF REAL-WORLD DATASETS

**Problem Setting.** The RRNCO dataset (Son et al., 2026) is used to evaluate the performance of neural solvers in real-world instances. It is constructed from 100 cities distributed across six continents: 25 in Asia, 21 in Europe, 15 in North America, 15 in South America, 14 in Africa, and 10 in Oceania. For each city, a 3×3 km region is extracted from OpenStreetMap (OpenStreetMap contributors, 2017). The travel distances and durations between locations are calculated using a locally deployed OSRM server (Luxen & Vetter, 2011). VRP instances are subsequently generated by subsampling locations and assigning demand vectors and time windows, while the original spatial topology and routing constraints are preserved.

Table 7: Comparasion with Matnet Augmentation on ATSP.

| Method | ATSP100 | | | ATSP200 | | | ATSP500 | | | ATSP1000 | | |
|---|---|---|---|---|---|---|---|---|---|---|---|---|
| | Obj. | Gap | Time | Obj. | Gap | Time | Obj. | Gap | Time | Obj. | Gap | Time |
| LKH-1000 | 1.5643 | * | 5.91m | 1.5721 | * | 16.40m | 1.5763 | * | 44.45m | 1.5739 | * | 1.71h |
| Matnet (×128) | 1.6002 | 2.29% | 4.39m | 3.0272 | 92.56% | 20.34m | – | – | – | – | – | – |
| Matnet$^+$ (×128) | 1.5863 | 1.41% | 4.40m | 1.8428 | 17.22% | 20.34m | – | – | – | – | – | – |
| Matnet-Single (One-hot)$^+$ (×128) | 1.5778 | 0.86% | 3.16m | 1.6305 | 3.71% | 16.33m | – | – | – | – | – | – |
| Matnet-Single (Random)$^+$ (×128) | 1.5780 | 0.88% | 3.14m | 1.6173 | 2.88% | 16.33m | 1.8061 | 14.58% | 2.95m | 2.0836 | 32.38% | 21.84h |
| RADAR$^+$ | **1.5756** | **0.72%** | 0.04m | **1.5879** | **1.01%** | 0.15m | **1.6098** | **2.13%** | 1.45m | **1.6389** | **4.13%** | 11.57m |

Note. $^+$ indicates training and testing with z-score normalization. The boldface indicates the best result among learning-based methods.

**Training Setting.** We follow the same training setting as RRNCO for a fair comparison. We train the model for 200 epochs using the Adam optimizer with an initial learning rate of $4 \times 10^{-4}$, which is decayed by a factor of 0.1 at epochs 180 and 195. Each epoch processes 100,000 instances with a batch size of 256. The model employs 128-dimensional node embeddings, 8 attention heads, 512-dimensional feedforward layers, and a total of 12 Transformer layers.

**Testing Setting.** We follow the same testing setting as RRNCO. The test batch size is set to 32. Data augmentation is made possible by the inclusion of coordinate features. Thus, we apply a data augmentation factor of 8 to the input coordinates. Each test set consists of 1280 instances of size 100, and the reported inference time is measured over the entire test set.

**Baseline.** Given that our experiments follow the same datasets and evaluation protocol as RRNCO, we directly adopt the baseline results reported in their work. The selected baselines include Mat-Net (Kwon et al., 2021) RRNCO, and GCN (Duan et al., 2020). Other models are not included in the comparison, mainly due to, 1) RRNCO already achieved state-of-the-art performance among neural solvers as reported in (Son et al., 2026), and 2) most of the baseline methods used in (Son et al., 2026), such as LEHD (Luo et al., 2023) and POMO (Kwon et al., 2020), do not support explicit distance matrices as input, making the comparison unfair.

**Our Model.** To verify the effectiveness of our method, we build upon the RRNCO framework by replacing its original distance embedding with our SVD-based initialization and modifying the attention mechanism to Matnet attention with Sinkhorn normalization.

# C MORE RESULTS AND DISCUSSION

## C.1 METNET AUGMENTATION

MatNet's one-hot node initialization introduces stochasticity in the constructive policy. To make use of this, the original paper introduces an augmentation scheme that repeats each instance 128 times and returns the best solution among the independent runs (we denote this as MatNet×128). In this section, we report MatNet_X128 on ATSP and also include new variants for comparison. For ACVRP, MatNet variants in our setup employ demand-based projections (deterministic), so the 128-repeat augmentation is not applicable. As indicates in Table 7, matNet augmentation reduces the gap but incurs a substantial runtime increase and still fails to surpass RADAR$^+$ across scales.

## C.2 MULTI-TASK LEARNING ON VARIOUS ASYMMETIC VRP VARIANTS

This section reports detailed multitask results across 16 asymmetric VRP variants. Across all problems, RADAR consistently surpasses the learning-based baselines, achieving the lowest gaps and strong overall efficiency; see Table 8 for per-variant numbers.

## C.3 DIFFERENT DEMAND DISTRIBUTION

In VRP models, customer demands are typically sampled from a discrete uniform distribution over $\{1, \ldots, 9\}$. While this demand setting is convenient for training and evaluation, it provides only a limited view of the demand landscape. A model that fits well under this single distribution may lack sensitivity to distributional shifts, leading to degraded performance once the demand law deviates from the training regime. To probe robustness and distribution awareness, we further evaluate it on two out-of-distribution demand distributions. **Skewed-small.** 80% of customer demands are sampled

Table 8: Performance on 16 VRP variants at problem size $N = 100$ (each with 1K test instances).

| Method | ACVRP Obj. | Gap | Time | AOVRP Obj. | Gap | Time | AVRPB Obj. | Gap | Time | AVRPL Obj. | Gap | Time |
|---|---|---|---|---|---|---|---|---|---|---|---|---|
| HGS | 2.197 | * | 5.93m | 1.587 | * | 5.81m | 2.285 | * | 5.90m | 2.191 | * | 6.21m |
| OR-Tools | 2.326 | 5.87% | 6.01m | 1.665 | 4.91% | 6.07m | 2.397 | 4.90% | 5.99m | 2.308 | 5.34% | 5.83m |
| RF | 2.246 | 2.23% | 2.39s | 1.635 | 3.02% | 1.98s | 2.363 | 3.41% | 1.89s | 2.242 | 2.37% | 1.91s |
| RF-NN | 2.232 | 1.59% | 3.11s | 1.626 | 2.46% | 1.99s | 2.348 | 2.76% | 1.90s | 2.229 | 1.73% | 1.94s |
| RADAR | **2.222** | **1.14%** | 3.51s | **1.618** | **1.95%** | 2.39s | **2.334** | **2.14%** | 2.31s | **2.218** | **1.23%** | 2.34s |

| Method | AOVRPB Obj. | Gap | Time | AOVRPL Obj. | Gap | Time | AVRPBL Obj. | Gap | Time | AVRPTW Obj. | Gap | Time |
|---|---|---|---|---|---|---|---|---|---|---|---|---|
| HGS | 1.763 | * | 5.90m | 1.594 | * | 6.03m | 2.286 | * | 5.89m | 3.281 | * | 5.87m |
| OR-Tools | 1.820 | 3.23% | 6.01m | 1.665 | 4.45% | 5.98m | 2.402 | 5.07% | 6.09m | 3.328 | 1.43% | 6.00m |
| RF | 1.804 | 2.33% | 2.00s | 1.643 | 3.07% | 1.99s | 2.364 | 3.41% | 1.92s | 3.386 | 3.20% | 1.96s |
| RF-NN | 1.797 | 1.93% | 2.02s | 1.634 | 2.51% | 2.00s | 2.351 | 2.76% | 1.92s | 3.374 | 2.83% | 1.98s |
| RADAR | **1.784** | **1.19%** | 2.43s | **1.625** | **1.94%** | 2.42s | **2.335** | **2.14%** | 2.35s | **3.344** | **1.92%** | 2.38s |

| Method | AOVRPBL Obj. | Gap | Time | AOVRPTW Obj. | Gap | Time | AVRPBTW Obj. | Gap | Time | AVRPLTW Obj. | Gap | Time |
|---|---|---|---|---|---|---|---|---|---|---|---|---|
| HGS | 1.760 | * | 6.03m | 2.443 | * | 5.87m | 3.581 | * | 5.83m | 3.277 | * | 5.93m |
| OR-Tools | 1.831 | 4.03% | 5.87m | 2.457 | 0.57% | 6.00m | 3.666 | 2.37% | 6.01m | 3.372 | 2.90% | 5.90m |
| RF | 1.804 | 2.50% | 2.01s | 2.467 | 0.98% | 2.12s | 3.699 | 3.30% | 1.99s | 3.388 | 3.39% | 2.00s |
| RF-NN | 1.795 | 1.99% | 2.02s | 2.459 | 0.65% | 2.00s | 3.681 | 2.79% | 1.92s | 3.374 | 2.96% | 1.98s |
| RADAR | **1.782** | **1.25%** | 2.43s | **2.444** | **0.00%** | 2.55s | **3.649** | **1.90%** | 2.41s | **3.350** | **2.23%** | 2.42s |

| Method | AOVRPBTW Obj. | Gap | Time | AOVRPLTW Obj. | Gap | Time | AVRPBLTW Obj. | Gap | Time | AOVRPBLTW Obj. | Gap | Time |
|---|---|---|---|---|---|---|---|---|---|---|---|---|
| HGS | 2.624 | * | 6.06m | 2.435 | * | 5.86m | 3.604 | * | 5.87m | 2.627 | * | 5.97m |
| OR-Tools | 2.653 | 1.11% | 5.87m | 2.462 | 1.11% | 6.00m | 3.667 | 1.75% | 6.01m | 2.636 | 0.34% | 6.05m |
| RF | 2.650 | 0.99% | 2.15s | 2.458 | 0.94% | 2.12s | 3.727 | 3.41% | 2.02s | 2.652 | 0.95% | 2.16s |
| RF-NN | 2.640 | 0.61% | 2.02s | 2.449 | 0.57% | 2.00s | 3.710 | 2.94% | 1.92s | 2.646 | 0.72% | 1.98s |
| RADAR | **2.627** | **0.11%** | 2.56s | **2.435** | **0.00%** | 2.52s | **3.676** | **2.00%** | 2.43s | **2.632** | **0.19%** | 2.57s |

**Note.** The boldface indicates the best result among learning-based methods.

Table 9: Performance under skewed demand distributions of ACVRP100.

| Method | Skewed-Large Obj. | Gap | Time | Skewed-Small Obj. | Gap | Time |
|---|---|---|---|---|---|---|
| LKH-100 | 2.3626 | 4.97% | 2.19m | 1.8474 | 2.05% | 2.04m |
| LKH-1000 | 2.2507 | – | 18.08m | 1.8105 | – | 17.55m |
| MatNet | 2.2806 | 1.33% | 0.04m | 1.8937 | 4.60% | 0.04m |
| MatNet-Single (Demand) | 2.2641 | 0.60% | 0.03m | 1.8821 | 3.95% | 0.03m |
| MatNet-Single (Random) | 2.2631 | 0.55% | 0.03m | 1.8760 | 3.63% | 0.03m |
| ReLD | 2.2480 | -0.12% | 0.04m | 1.8575 | 2.60% | 0.04m |
| RADAR | **2.2428** | **-0.35%** | 0.04m | **1.8536** | **2.38%** | 0.03m |

**Note.** The boldface indicates the best result among learning-based methods.

from $\mathcal{U}(1, 3)$, and the remaining 20% from $\mathcal{U}(4, 10)$. **Skewed-large.** 80% of customer demands are sampled from $\mathcal{U}(4, 10)$, and the remaining 20% from $\mathcal{U}(1, 3)$. As shown in Table 9, our method consistently attains the lowest average cost under both skewed distributions.

# D  ABLATION STUDY

## D.1  CURRENT INITIALIZATION METHODS

Figure 2 shows the generalization performance of different initialization methods trained on ATSP100 and evaluated on larger sizes. MatNet and UNICO use dual embeddings, while others use single embeddings. ICAM includes distances to the 50 nearest neighbors, and RRNCO samples 25 neighbors based on distance-derived probabilities.

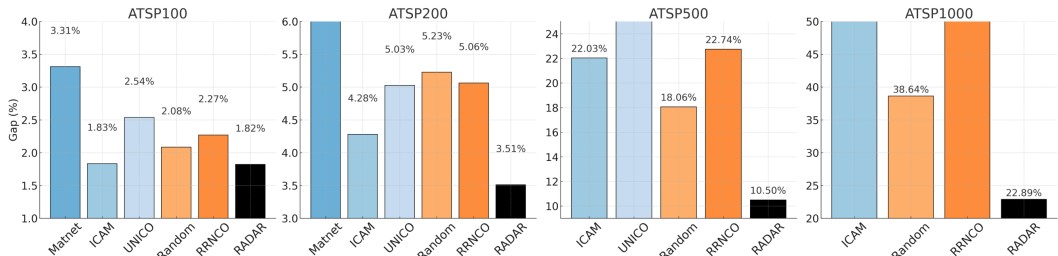

Figure 2: Different initialization performance under varying size of ATSP.

The figure shows that the performance differences between initialization methods become more pronounced as the problem size increases. While several methods exhibit significant degradation, our initialization remains stable across scales. This indicates that a well-designed initialization improves solution quality on large instances and contributes to better generalization across problem sizes.

### D.2 ALTERNATIVE SVD METHODS

To substantiate the effectiveness of our SVD-based embedding, we compare it against several common matrix–factorization baselines. Below we summarize the intuition behind each family and why their behavior differs in our setting.

**Eigenvalue Decomposition (EVD).** EVD factors matrix as $Q\Lambda Q^\top$. It captures global spectral structure well on symmetric distances. Analogous to our truncated setup, we retain the top-$k$ eigenpairs (here $k{=}10$). Because for symmetric decompositions the left and right eigenvectors coincide, we form the embedding with a single factor $Q_k$ and $\Lambda_k^{1/2}$.

**Multidimensional Scaling (MDS).** Classical MDS applies double–centering $B = -\frac{1}{2}JD^2J$ with $J = I - \frac{1}{n}\mathbf{1}\mathbf{1}^\top$, and recovers an embedding by the EVD of $B$. Compared with plain EVD, MDS improves the geometry by (i) removing the effect of global translation via $J$, and (ii) operating on squared distances so that $B$ behaves like a Gram matrix of an Euclidean embedding.

**QR Decomposition.** QR factorizes $A = QR$ with $Q$ orthonormal and $R$ upper–triangular. Since QR decomposition provides no spectral ordering, we take the first $k$ columns of $Q$ and the first $k$ rows of $R$ and use their product to go through projection layer to obtain the embeddings.

**Random Approximation (RA).** We uniformly initialize a low-dimensional vector for each node. To encourage asymmetry-aware embeddings, we add an *information loss* that penalizes the mismatch between the predicted scores and $D$:

$$\mathcal{L}_{\text{info}} \;=\; \big\| XW_1\big(XW_2\big)^\top - D \big\|_1, \tag{7}$$

where we use an elementwise $\ell_1$ distance. The overall objective is

$$\mathcal{L}_{\text{total}} \;=\; \mathcal{L}_{\text{task}} \;+\; \lambda\,\mathcal{L}_{\text{info}}, \tag{8}$$

with $\lambda > 0$ balancing task performance and the asymmetry-aware constraint.

From the results in Table 5, several key observations can be made.

**Eigenvalue-based methods.** EVD and MDS consistently achieve competitive results across different problem sizes. Their advantage lies in capturing global spectral structures, which are beneficial for encoding relational information. However, since EVD is fundamentally designed for symmetric matrices, its direct application to asymmetric distance matrices leads to degraded performance, as reflected in the moderate gap compared to the RADAR baseline.

**QR decomposition.** In contrast, the QR-based approach yields the weakest performance across all scales. Unlike EVD, QR decomposition does not produce eigenvalues and therefore lacks a natural criterion for ordering feature significance. When truncated to a fixed dimension, this absence of a ranking mechanism results in substantial structural loss.

Table 10: Comparison with other SVD-based methods.

| Method | ATSP100 | | ATSP200 | | ATSP500 | | ATSP1000 | |
|--------|---------|------|---------|------|---------|------|----------|-------|
| | Obj. | Time | Obj. | Time | Obj. | Time | Obj. | Time |
| EVD | 1.5960 | 0.04 | 1.6420 | 0.16 | 1.8447 | 1.47 | 2.2576 | 11.81 |
| QR | 1.6137 | 0.03 | 1.6822 | 0.13 | 2.0650 | 1.34 | 2.7875 | 11.47 |
| MDS | 1.5983 | 0.05 | 1.6371 | 0.19 | 1.7893 | 1.59 | 2.0737 | 12.05 |
| RA1-L1 | 1.6003 | 0.03 | 1.6577 | 0.12 | 1.9356 | 1.30 | 2.5125 | 11.11 |
| RA2-L2 | 1.5971 | 0.03 | 1.6448 | 0.13 | 1.8384 | 1.30 | 2.2518 | 11.11 |
| RA1-L5 | 1.5973 | 0.03 | 1.6548 | 0.13 | 1.9087 | 1.30 | 2.3916 | 11.11 |
| RADAR | **1.5928** | 0.04 | **1.6273** | 0.15 | **1.7418** | 1.45 | **1.9342** | 11.57 |

**Note.** RA1-L1 denotes injecting the information loss after the first encoder layer with $\lambda = 1$; RA2-L2 uses the same placement with $\lambda = 2$; RA1-L5 injects the information loss after the fifth encoder layer with $\lambda = 1$.

**Random Approximation (RA) variants.** The RA family shows partial improvements when stronger geometric guidance is injected, either by increasing the loss coefficient or by pushing the injection point deeper into the encoder. Nevertheless, the overall performance of the RA variants remains inferior to RADAR. This suggests that the information loss supervision, while theoretically encouraging asymmetry awareness, also increases the optimization burden.

### D.3 EFFECT OF $k$ IN INFORMED EMBEDDING.

Figure 3 illustrates the top-$k$ selection distribution across different instance sizes for three informed initialization methods: ICAM, RRNCO, and RADAR. Each subplot corresponds to a different top-$k$ threshold (10, 30, 50), showing how frequently each method selects high-quality solutions at different ATSP problem sizes (100, 200, 500, 1000). The radar plots provide a visual summary of how each method behaves under varying $k$ values and problem scales. The efficiency score is defined as $1 - \text{GAP}$, where higher values indicate better initialization quality.

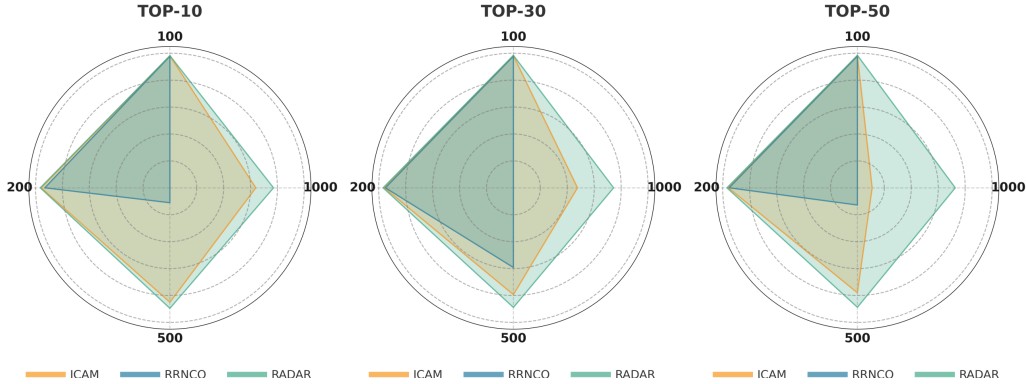

Figure 3: Efficiency Score of informed initialization across different top k on ATSP.

We can observe that as $k$ increases from 10 to 50, a general decline in performance is observed across all methods, suggesting a reduced generalization under larger candidate sets. RADAR consistently outperforms ICAM and RRNCO across all settings, with the performance difference becoming more pronounced as the problem size increases. Based on this trade-off between candidate flexibility and generalization performance, we adopt $k = 10$ as the default setting in our experiments.

### D.4 SVD RUNTIME ANALYSIS

To quantify the runtime impact of the SVD step, we conduct two complementary experiments.

**Remove SVD.** We retrain a variant with the SVD step removed and compare end-to-end training time and inference latency against full RADAR. The ablation shows a modest reduction in training

| Method | Train Time | ATSP100 | ATSP200 | ATSP500 | ATSP1000 |
|--------|-----------|---------|---------|---------|----------|
| RADAR | 39.31h | 0.04m | 0.15m | 1.45m | 11.57m |
| w/o SVD | 35.67h | 0.03m | 0.13m | 1.43m | 11.37m |

Table 11: Ablation on SVD. Removing SVD shortens training by 9% with negligible differences in inference latency across sizes.

time, while inference latencies across ATSP-100/200/500/1000 remain essentially unchanged. These results indicate that SVD is not the main runtime bottleneck at scale.

**Profiling of the SVD step.** We profiled the runtime overhead of SVD during inference. Specifically, we reused the RADAR test setup and evaluated 1,000 instances per setting across multiple problem sizes. As shown in Fig. 4, while the total runtime increases with problem size, the relative cost of SVD diminishes and is no longer the dominant components at larger scales.

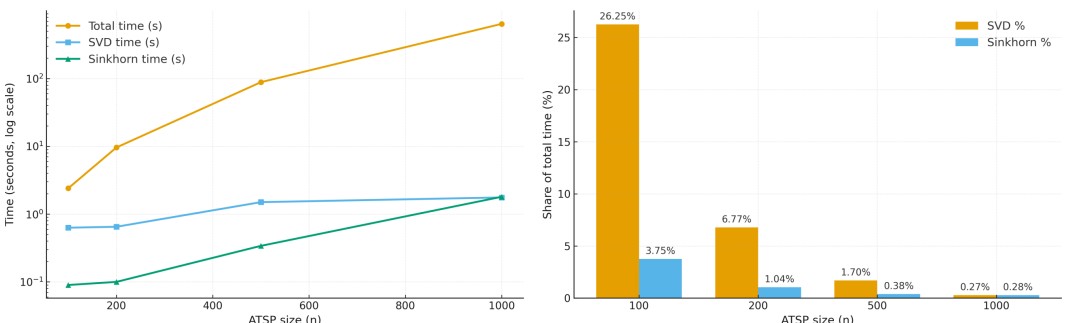

Figure 4: The left panel shows wall-clock time versus ATSP size (log scale) for the total pipeline, SVD, and Sinkhorn. The right panel reports the percentage share of the total runtime attributable to SVD and Sinkhorn at each size. Each measurement averages 1,000 test instances.

Finally, considering that many real world routing problems may often involve fewer than one thousand nodes owing to logistical, geographical or operational limitations, we believe that the computational burden of current SVD step remains well within acceptable bounds for practical applications.

### D.5 SINKHORN V.S SOFTMAX

To test whether Sinkhorn can serve as a drop-in alternative to softmax, we compared the first 10 epochs and the final 10 epochs (2091–2100) on ATSP100 under identical settings except for the attention normalization. Sinkhorn reduces the score from 3.06 to 1.69 by epoch 10, whereas softmax goes from 3.05 to 1.79. In late training, Sinkhorn stabilizes around 1.577–1.579, while softmax remains around 1.591–1.596. Thus, Sinkhorn both converges faster early on and attains a lower final score, supporting its use as a substitute for softmax in ATSP.

### D.6 SINKHORN RUNTIME ANALYSIS

We measured the cumulative runtime of SINKHORN across ATSP test sets of increasing size (see Fig. 4). Empirically, its share of the overall wall-clock time tends to decrease as problem size grows and remains modest at the largest size we evaluate. This suggests that, under our implementation, the normalization step scales favorably and is unlikely to dominate end-to-end runtime.

### D.7 NUMBER OF SINKHORN ITERATIONS

We additionally conducted a sensitivity study on the ATSP with the number of Sinkhorn iterations set to $T \in \{1, 5, 10\}$. Table 12 indicates that increasing $T$ consistently improves solution quality, while the associated runtime overhead remains marginal. Therefore, a larger $T$ yields clear accuracy gains with almost no additional computational cost.

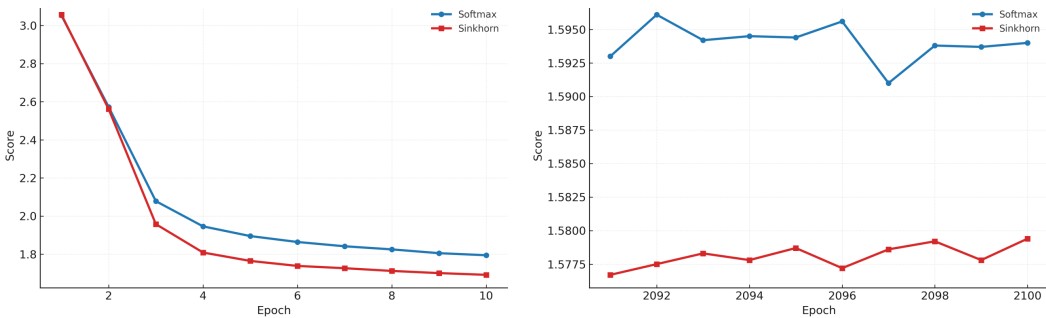

Figure 5: ATSP-100: Softmax vs. Sinkhorn. Left: Early training (epochs 1–10) showing faster convergence with Sinkhorn. Right: Late training (epochs 2091–2100) zoo m-in, where Sinkhorn stabilizes at a lower objective than Softmax.

Table 12: Effect of Sinkhorn Iterations on RADAR.

| Method | Iteration | 100 | GAP | Time | 200 | GAP | Time | 500 | GAP | Time | 1000 | GAP | Time |
|--------|-----------|-----|-----|------|-----|-----|------|-----|-----|------|------|-----|------|
| RADAR | 1 | 1.5815 | 1.10% | 0.04m | 1.6005 | 1.81% | 0.14m | 1.6542 | 4.94% | 1.43m | 1.7576 | 11.67% | 11.42m |
| RADAR | 5 | 1.5764 | 0.77% | 0.04m | 1.5909 | 1.20% | 0.14m | 1.6210 | 2.84% | 1.44m | 1.6590 | 5.41% | 11.50m |
| RADAR | 10 | **1.5756** | **0.72%** | 0.04m | **1.5879** | **1.01%** | 0.15m | **1.6098** | **2.13%** | 1.45m | **1.6389** | **4.13%** | 11.57m |

## D.8 STATISTICAL SIGNIFICANCE EXPERIMENT

To validate the training stability and performance advantage of our model, we conduct a statistical significance experiment based on multiple independent runs. We select ReLD (Huang et al., 2025), the strongest baseline model, as the comparison target. For each model, we run three independent training processes on the ATSP100 dataset and record the training scores at each epoch.

Figure 6 shows the training curves of RADAR and ReLD on ATSP100. The shaded regions denote the standard deviation over multiple runs, indicating variance across training trajectories. The left panel corresponds to the first 10 epochs (early stage), and the right panel corresponds to the final 10 epochs (late stage), highlighting differences in both convergence behavior and final stability.

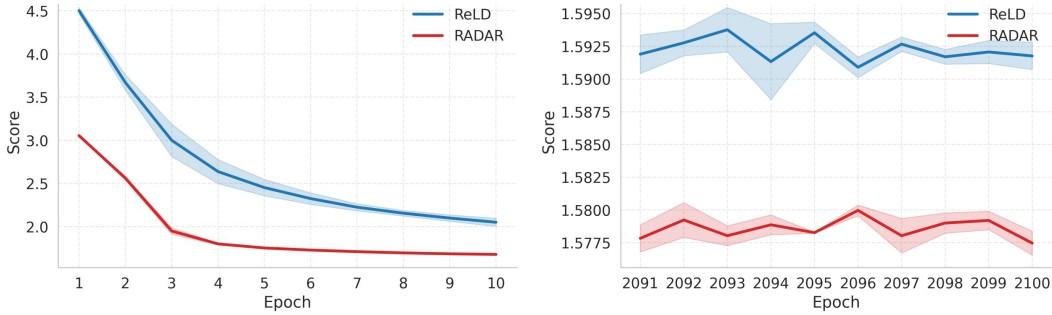

Figure 6: Training curves of RADAR and ReLD on ATSP100: early-stage (left) and late-stage (right) comparison over the first and last 10 epochs.

RADAR achieves lower scores than ReLD in both stages. Moreover, its training curves exhibit a smaller variance, as indicated by the narrower shaded regions. This suggests that RADAR not only converges faster, but also maintains more stable performance with less fluctuation during training.

To further validate the robustness of our approach, we compare the test performance of RADAR and ReLD. Figure 7 shows the score distributions of RADAR and ReLD on four ATSP test sets with 100, 200, 500, and 1000 nodes, respectively. Each boxplot summarizes the results over 1000 distinct test instances. Statistical significance is assessed using two sample $t$-tests, with significance levels denoted by asterisks: $*p < 0.05$, $**p < 0.01$, and $***p < 0.001$.

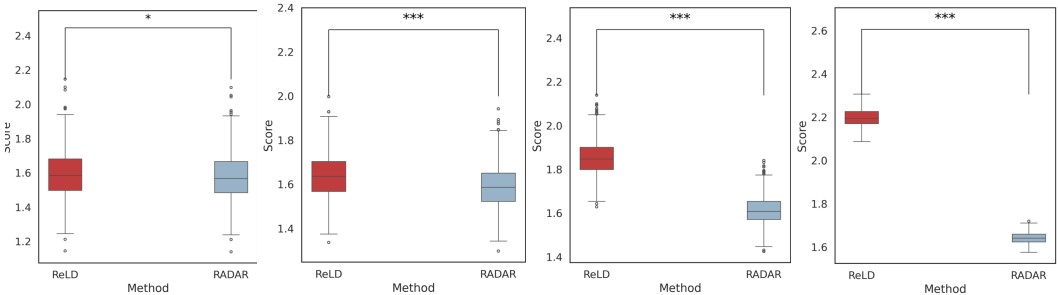

Figure 7: Statistical comparison of RADAR and ReLD on ATSP test instances. Each subplot presents the score distributions over multiple runs on ATSP100, ATSP200, ATSP500, and ATSP1000 test sets, respectively. Statistical significance is assessed using independent two-sample $t$-tests. Asterisks indicate significance levels: $*p < 0.05$, $**p < 0.01$, and $***p < 0.001$.

Table 13: Performance of Different Normalization on Synthetic ATSP.

| Method | 100 | GAP | Time | 200 | GAP | Time | 500 | GAP | Time | 1000 | GAP | Time |
|---|---|---|---|---|---|---|---|---|---|---|---|---|
| LKH | 1.5643 | – | 1.05m | 1.5721 | – | 4.43m | 1.5763 | – | 29.96m | 1.5739 | – | 130.15m |
| RADAR- | 1.5840 | 1.26% | 0.04m | 1.6939 | 7.75% | 0.15m | 3.7412 | 137.34% | 1.45m | 6.0139 | 281.42% | 11.57m |
| RADAR+ | 1.5776 | 0.85% | 0.04m | 1.5917 | 1.25% | 0.15m | 1.6247 | 3.07% | 1.45m | 1.6756 | 6.27% | 11.57m |
| RADAR* | **1.5756** | **0.72%** | 0.04m | **1.5879** | **1.01%** | 0.15m | **1.6098** | **2.13%** | 1.45m | **1.6389** | **4.13%** | 11.57m |
| ReLD- | diverge | – | – | diverge | – | – | diverge | – | – | diverge | – | – |
| ReLD+ | 1.5976 | 2.13% | 0.03m | 1.6403 | 4.34% | 0.15m | 1.8207 | 15.50% | 1.49m | 2.4648 | 56.32% | 12.18m |
| ReLD* | **1.5900** | **1.64%** | 0.03m | **1.6310** | **3.75%** | 0.15m | **1.7873** | **13.39%** | 1.49m | **2.0723** | **31.67%** | 12.18m |
| Matnet-Single (Random)- | diverge | – | – | diverge | – | – | diverge | – | – | diverge | – | – |
| Matnet-Single (Random)+ | 1.6245 | 3.85% | 0.02m | 1.6907 | 7.54% | 0.13m | 1.9651 | 24.67% | 1.34m | 2.8334 | 79.70% | 11.11m |
| Matnet-Single (Random)* | **1.5969** | **2.08%** | 0.02m | **1.6543** | **5.23%** | 0.13m | **1.8610** | **18.10%** | 1.34m | **2.1821** | **38.64%** | 11.11m |
| ICAM- | **1.6440** | **5.09%** | 0.01m | 1.9715 | 25.41% | 0.06m | 2.6485 | 68.02% | 0.73m | 3.3871 | 114.82% | 9.80m |
| ICAM+ | 1.6638 | 6.36% | 0.01m | 1.9735 | 25.53% | 0.06m | 3.0408 | 92.91% | 0.73m | 4.3716 | 177.26% | 9.80m |
| ICAM* | 1.6580 | 5.99% | 0.01m | **1.8471** | **17.49%** | 0.06m | **2.4592** | **56.01%** | 0.73m | **2.9069** | **84.69%** | 9.80m |
| One_hot- | diverge | – | – | diverge | – | – | diverge | – | – | diverge | – | – |
| One_hot+ | 1.6038 | 2.53% | 0.02m | **1.6601** | **5.60%** | 0.12m | – | – | – | – | – | – |
| One_hot* | **1.5995** | **2.25%** | 0.02m | 1.6727 | 6.40% | 0.12m | – | – | – | – | – | – |
| Matnet- | 1.6473 | 5.31% | 0.03m | 3.1925 | 103.07% | 0.14m | – | – | – | – | – | – |
| Matnet+ | **1.6158** | **3.29%** | 0.03m | 3.1493 | 100.32% | 0.14m | – | – | – | – | – | – |
| Matnet* | **1.6158** | **3.29%** | 0.03m | **1.9111** | **21.56%** | 0.14m | – | – | – | – | – | – |
| ELG- | 1.6080 | 2.79% | 0.06m | 1.8492 | 17.63% | 0.27m | 2.3966 | 52.04% | 2.60m | 2.5286 | 60.37% | 22.31m |
| ELG+ | **1.5907** | **1.69%** | 0.06m | **1.6287** | **3.60%** | 0.27m | **1.7404** | **10.41%** | 2.60m | 1.9380 | 22.91% | 22.31m |
| ELG* | 1.5982 | 2.17% | 0.06m | 1.6423 | 4.47% | 0.27m | 1.7456 | 10.74% | 2.60m | **1.8441** | **17.17%** | 22.31m |

**Note.** the suffix "-" denotes no normalization, "+" denotes min–max normalization, and "*" denotes z-score normalization. Boldface indicates the best-performing normalization variant within each method.

RADAR consistently produces lower scores than ReLD across all instance sizes. The performance gap increases with problem scale, and the statistical significance of the difference becomes more evident on larger instances, reflecting RADAR's stronger scalability and effectiveness in asymmetric settings.

## E    EFFECT OF NORMALIZATION

The use of z-score normalization in Table 1 was guided by a standard deep-learning consideration: normalizing input scales reduces sensitivity to magnitude variations and stabilizes gradient flow. This is especially important for attention-based models that consume raw distance matrices, whose values can differ substantially across instances. To verify that imposing a uniform z-score setting on all baselines does not bias the comparison, we re-evaluated all constructive neural solvers on the ATSP benchmarks under three strictly matched input regimes (z-score, min–max normalization, and no normalization), while keeping architectures, training schedules, and evaluation protocols identical. Table 13 indicates z-score normalization remained consistently competitive and, in most cases, yielded the best performance relative to min–max or unnormalized inputs.

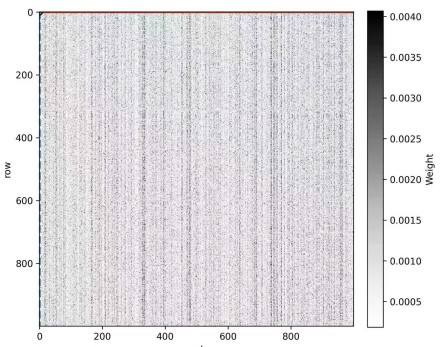 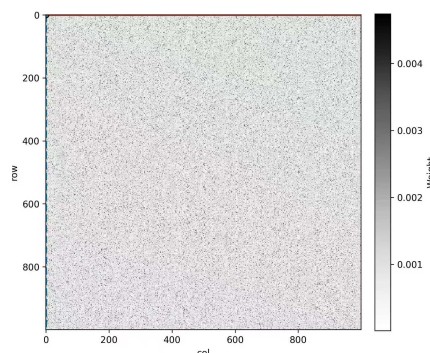

(a) Softmax. Clear attractor patterns.      (b) Sinkhorn. Uniformly distributed weight map.

Figure 8: Visualization of attention weight matrices produced by Softmax and Sinkhorn.

Table 14: Infeasible rate on ACVRP instances.

| Infeasible Rate | ACVRP100 | ACVRP200 | ACVRP500 | ACVRP1000 |
|---|---|---|---|---|
| **HGS-Short** | 25.9% | 48.7% | 92.2% | 6.6% |
| **HGS-Long** | 2.5% | 0.1% | 4.6% | 0.00% |

## F    Sinkhorn Attention Heatmap

Sinkhorn normalization improves generalization because it introduces an explicit column-communication mechanism. With standard Softmax attention, each row is normalized independently, so the model can assign disproportionately large probability mass to a few "attractor" nodes that naturally receive high incoming attention from many rows. This creates strong column imbalance: a small set of nodes dominate the attention matrix, while most columns stay uniformly low. During decoding, once those high-attention nodes are visited and removed from the candidate set, the remaining columns are all similarly small, so the attention distribution becomes nearly flat and the model is forced to make decisions under high uncertainty because there is no clear preference signal left. Sinkhorn iteratively normalizes both rows and columns. This prevents any single column from absorbing too much global mass and redistributes attention more evenly across nodes. As a result, even after some nodes are selected, the remaining nodes still maintain meaningful relative attention differences instead of collapsing to a uniform low baseline. In practice, this makes the decoder's selection policy more stable and less dependent on distribution-specific "hub" patterns.

## G    Infeasible Rate of HGS on Synthetic ACVRP

Since HGS can return infeasible solutions, we additionally report the infeasible rate to provide a more complete picture of its practical reliability under each computational budget. The infeasible rate denotes the proportion of runs in which HGS fails to construct a solution that satisfies all vehicle capacity constraints. As shown in Table 14, the short-budget configuration (HGS-Short) exhibits a substantial fraction of infeasible runs. Increasing the time budget to HGS-Long substantially improves robustness, driving the infeasible rate down to 2.5% on ACVRP100, below 5% on ACVRP500, and effectively zero for ACVRP200 and ACVRP1000. These results highlight that, while HGS can be made highly reliable given sufficient runtime, its solution quality under tight time budgets must be interpreted together with the corresponding infeasible rate.

## H    More Scenarios on SVD-based Initialization

Our approach can apply to a broader class of tasks where the instance can be represented by a **relational matrix** $R$ (not necessarily distance matrix). $R$ encodes directed or asymmetric relations between entities, such as costs, compatibilities, precedences, processing times, or transition penalties.

Table 15: Results on FFSP with and without SVD-based initialization.

| Method | FFSP20 | | | FFSP50 | | | FFSP100 | | |
|---|---|---|---|---|---|---|---|---|---|
| | MS | Gap | Time | MS | Gap | Time | MS | Gap | Time |
| MatNet | 28.0676 | 0.3 | 0.82m | 52.5973 | 0.6 | 1.36m | 93.0295 | 0.6 | 2.39m |
| MatNet_SVD | **27.9727** | **0.0** | 0.85m | **52.3042** | **0.0** | 1.39m | **92.4473** | **0.0** | 2.43m |

By decomposing $R$ into asymmetric components at initialization and injecting these components into the attention mechanism, the model can recover and propagate relational signals throughout message passing, rather than relying solely on node-wise features.

A representative example beyond routing is the Flexible Flow Shop Scheduling Problem (FFSP). In FFSP, multiple production stages each contain several parallel machines, and each job must be processed sequentially across stages. The decision is to assign jobs to machines and determine the processing order to minimize makespan or total completion time. In MatNet, FFSP instances are encoded as matrices of processing times between jobs and machines at each stage. These entries are relational features rather than geometric distances: they describe directed interactions such as "job $i$ processed on machine $m$ requires time $p_{i,m}$".

To illustrate the generality of our method in this setting, we applied the proposed asymmetric SVD-based relational-feature encoding to MatNet under the FFSP configuration. Specifically, we replaced MatNet's original initialization with our decomposition-based initialization and retrained the model on instances of size $N = 20$. We then evaluated the resulting model in a zero-shot fashion on larger out-of-distribution sizes $N = 50$ and $N = 100$. The results (Table 15) show that the same decomposition-plus-attention mechanism extends naturally to non-routing relational problems and yields strong generalization beyond the training size.

We report MS (makespan)as the primary objective, which measures the completion time of the last job in the schedule; lower MS indicates a better schedule with shorter overall production horizon. Under this greedy inference setting, replacing MatNet's original matrix encoder with our SVD-based relational-feature encoding yields consistent improvements across sizes.

# I LARGE LANGUAGE MODELS (LLMS) USAGE

We employed LLMs in a limited, non–substantive role for (i) linguistic refinement and (ii) minor coding assistance.

**Writing.** All technical content and arguments were authored by the researchers. Drafts were subsequently polished with LLMs to improve clarity, grammar, and stylistic consistency; no conceptual or methodological changes were introduced by LLM outputs.

**Coding.** Implementation work was primarily derived from Matnet (Kwon et al., 2021), and the design of our proposed SVD-based initialization and Sinkhorn normalization was developed independently by the authors.

**Scope and provenance.** LLM-assisted edits were performed under human supervision and verified for correctness. All ideas, algorithms, and results are attributable to the authors; LLMs were not used for data generation, evaluation, or decision making.

