# OpenReview forum: "RADAR: Learning to Route with Asymmetry-aware Distance Representations"
_ICLR.cc/2026/Conference — ICLR 2026 Poster_

### Official Review · Reviewer_D1Sv · 2025-10-29

**Soundness:** 2
**Presentation:** 3
**Contribution:** 3
**Rating:** 4
**Confidence:** 3

**Summary:**

This paper introduces RADAR, a scalable neural framework designed to address the limitation of existing neural solvers for Vehicle Routing Problems (VRPs) that primarily rely on symmetric Euclidean distances, which restricts their applicability to real-world scenarios with asymmetric distances (e.g., one-way streets, traffic congestion).
RADAR tackles asymmetry from both static and dynamic perspectives: for static asymmetry, it uses Truncated Singular Value Decomposition (TSVD) on the asymmetric distance matrix to initialize compact, generalizable node embeddings that capture each node’s role as a source (outbound costs) and destination (inbound costs); for dynamic asymmetry, it replaces standard row-wise softmax normalization with Sinkhorn normalization, which jointly normalizes rows and columns of the attention matrix to enforce balanced bidirectional flow and capture global distance information from both interacting nodes.
Extensive experiments on 17 synthetic VRP variants (e.g., ATSP, ACVRP) and 3 real-world datasets show that RADAR outperforms SOTA traditional and neural baselines.
I regard the asymmetric vehicle routing problem as highly significant yet relatively underexplored in current research. The approach proposed in this paper is both novel and interesting; however, its experimental design and results are not very solid. Specifically, the selected baseline algorithms are not specifically tailored for asymmetric vehicle routing problems, limiting the validity of comparative analyses. Additionally, the experimental evaluations only cover out-of-distribution scenarios—where models are trained on instances of size 100 and tested on instances of other sizes—while failing to include in-distribution scenarios, i.e., training on instances across all sizes and subsequent testing on instances of all sizes. This omission undermines the comprehensiveness of the model’s performance verification.

**Strengths:**

1. The asymmetric vehicle routing problem is not well studied yet, this paper addresses a critical real-world gap by explicitly modeling asymmetric distances (e.g., one-way streets, traffic in vehicle routing problems (VRPs).
2. Features a dual-component design with strong theoretical grounding: Truncated Singular Value Decomposition (TSVD) initializes node embeddings to capture static asymmetry (distinguishing nodes as sources/destinations), while Sinkhorn normalization replaces softmax to model dynamic asymmetry via bidirectional global attention.
3. Demonstrates robust generalization across scenarios: it outperforms both traditional solvers (e.g., LKH3, HGS) and SOTA neural baselines (e.g., ReLD, MatNet) in synthetic multi-task (16 VRP variants) datesets and real-world datasets.
4. Well writing and easy to understand.

**Weaknesses:**

1. The ACVRP problem has been proposed for a long time, and there have been some dedicated efforts to solve this problem, including heuristic algorithms (e.g. Vigo D. A heuristic algorithm for the asymmetric capacitated vehicle routing problem[J]. European Journal of Operational Research, 1996, 89(1): 108-126.) and meta-heuristic algorithms (e.g. Leggieri V, Haouari M. A matheuristic for the asymmetric capacitated vehicle routing problem[J]. Discrete Applied Mathematics, 2018, 234: 139-150.). This paper does not discuss these previous works.
2. The selected baseline algorithms are not specifically tailored for asymmetric vehicle routing problems, limiting the validity of comparative analyses.
3. The experimental evaluations only cover out-of-distribution scenarios—where models are trained on instances of size 100 and tested on instances of other sizes—while failing to include in-distribution scenarios, i.e., training on instances across all sizes and subsequent testing on instances of all sizes, weakening result comparability.
4. It lacks an online update mechanism to adapt to real-time dynamic changes (e.g., sudden traffic jams).
5. The results of the ACVRPTW problem in Table 3. The method presented in this paper is not the optimal one, but it is highlighted in bold.

**Questions:**

I like the asymmetric vehicle routing problem and the idea of this article, but the current experimental results cannot fully demonstrate the effectiveness of this idea. If the author could provide additional explanations regarding the effectiveness of the idea, I would be happy to raise my rating score.
1. It is suggested to conduct comparative experiments on algorithms specifically designed to solve the ACVRP problem(e.g. heuristics and meta-heuristics).
2. The experimental design should incorporate in-distribution scenarios, specifically involving training the model on instances spanning all node sizes and subsequently evaluating its performance on test instances of all sizes.
3. The results in Table 5 indicate that the existing baseline algorithms are designed for symmetric structure problems and its performance will deteriorate in the case of asymmetry. Therefore, maybe the authors could compare the experimental results of the algorithm proposed in this paper and the baseline algorithm in the symmetric structure scenario?
4. Regarding the ATSP and ACVRP problems, maybe the authors could add the experimental results of the HGS algorithm?

---

> ### Author Response · Authors · 2025-11-26
> **Response (1/4)**
>
> >  **W1: The ACVRP problem has been proposed for a long time, and there have been some dedicated efforts to solve this problem, including heuristic algorithms (e.g. Vigo D. A heuristic algorithm for the asymmetric capacitated vehicle routing problem[J]. European Journal of Operational Research, 1996, 89(1): 108-126.) and meta-heuristic algorithms (e.g. Leggieri V, Haouari M. A matheuristic for the asymmetric capacitated vehicle routing problem[J]. Discrete Applied Mathematics, 2018, 234: 139-150.). This paper does not discuss these previous works.**
>
> Thank you for the reminder. We agree that ACVRP has a long OR history, with strong dedicated heuristics and matheuristics. Our motivation and claims, however, are specifically about NCO: rather than proposing yet another classical solver for ACVRP, we aim to address a current bottleneck in NCO, namely, that modern neural routing models struggle to preserve and exploit asymmetric relational structure when moving from symmetric benchmarks to realistic directed settings. So outperforming specialized OR heuristics is not a central claim of the paper.
>
> However, we are also happy to run and report these classical baselines. We want to note that the official codes for these ACVRP heuristics/matheuristics are not publicly available. Therefore, we made our best effort to re-implement the Vigo-style heuristic and the Leggieri-style matheuristic as faithfully as possible based on the papers.
>
> For algorithm Vigo D, we use Vigo’s parametric savings-based Clarke–Wright construction to rank directed customer pairs under asymmetry, then performs directed route merging with four endpoint/reversal cases. Then, the construction is permissive. It may accept merges that temporarily exceed capacity, controlled here by a penalty-based acceptance rule, and then restores feasibility using a cheapest-insertion repair stage. It also mirrors Vigo’s parameter exploration and multi-start strategy by searching a grid of savings parameters and adding small savings noise across restarts to escape bad constructions.
>
> For algorithm Leggieri, we first perform directed arc reduction to keep only the most promising arcs in the asymmetric graph, then solve a compact MILP seed on this reduced graph to obtain a high-quality feasible starting solution. After that, it follows their related-routes large-neighborhood strategy, repeatedly selecting a small set of mutually related routes and re-optimizing them via a sub-MILP, before reinserting the improved subsolution into the global routes. The implementation also keeps the paper’s emphasis on asymmetry-aware feasibility through flow-based capacity enforcement and variable vehicle-count search around the minimum required fleet size.
>
> |Method|100|Gap|Time|200|Gap|Time|500|Gap|Time|1000|Gap|Time|
> |-|-|-|-|-|-|-|-|-|-|-|-|-|
> |LKH-10000|2.1240|-|2.79h|2.1645|-|4.25h|2.3405|-|10.29h|2.4634|-|36.25h|
> |RADAR|2.1588|1.64%|0.05m|2.1483|-0.75%|0.14m|2.4198|3.39%|1.75m|2.4634|0.96%|11.60m|
> |VigoD|2.1539|1.41%|105.43h|-|-|-|-|-|-|-|-|-|
> |LeggieriV|2.1273|0.16%|4.44h|2.2802|5.35%|7.13h|-|-|-|-|-|-|
>
> >  **W2: The selected baseline algorithms are not specifically tailored for asymmetric vehicle routing problems, limiting the validity of comparative analyses.**
>
> We acknowledge this concern. At present, neural solvers for ACVRP/ATSP-style asymmetry are indeed limited, and there are very few publicly available models that are explicitly designed only for asymmetric VRP. Therefore, our baseline set is drawn from the strongest existing neural routing solvers that (i) include evaluation on asymmetric settings in their papers, and (ii) report improvements over MatNet under comparable asymmetric protocols. In other words, while these baselines may not be originally proposed only for asymmetric vehicle routing problems, they are precisely the neural methods that claim state-of-the-art performance for asymmetric vehicle routing problems.

---

> ### Author Response · Authors · 2025-11-26
> **Response (2/4)**
>
> >  **W3: The experimental evaluations only cover out-of-distribution scenarios—where models are trained on instances of size 100 and tested on instances of other sizes—while failing to include in-distribution scenarios, i.e., training on instances across all sizes and subsequent testing on instances of all sizes, weakening result comparability.**
>
> We respectfully clarify that our main training regime is size-100, so testing on N=100 is in-distribution. In neural combinatorial optimization, cross-size generalization is a central objective, so we additionally report three out-of-distribution settings to evaluate scalability.
>
> We agree that a mixed-size training evaluation is valuable for completeness, and we are happy to provide it. However, training on the full 100–1000 cross-size range is extremely compute-heavy: with our current hardware (RTX 3090), it would require on the order of 600+ hours to reproduce faithfully. As a practical compromise, we trained a mix-size model on sizes [50, 250] and then tested on N=100, 200, 500. Under this protocol, N=100 and N=200 are ID, while N=300 is OOD.
>
> For the mixed-size training experiment, we follow the ICAM protocol [1].
>
> - In each batch, we uniformly sample an instance size N from 50 to 250.
> - The batch size is scaled as:
>   `batch_size = 64 * (100 / N)^2`
> - Each epoch contains a fixed 1000 batches.
>
> We observed that training our original model in the mixed-size setting performs worse, mainly because it relies on Instance Normalization, whose per-instance statistics change with varying N. Replacing IN with Layer Normalization significantly stabilizes mix-size training and improves both in-distribution and out-of-distribution performance.
>
> |Method|100|Gap|200|Gap|500|Gap|
> |-|-|-|-|-|-|-|
> |RADAR|1.5756|0.72%|1.5879|1.01%|1.6098|2.13%|
> |RADAR-mix-IN|1.5877|1.50%|1.6056|2.13%|1.6365|3.82%|
> |RADAR-mix-LN|1.5766|0.79%|1.5864|0.91%|1.5942|1.14%|
>
> It can be seen that mixed-size training slightly sacrifices performance on small-size instances, but yields better results on large-size instances and improves cross-size generalization overall.
>
> >  **W4: It lacks an online update mechanism to adapt to real-time dynamic changes.**
>
> We sincerely thank your insightful question regarding dynamic asymmetry. In the literature, these problems are often formulated as time-dependent VRPs (TDVRPs). There are several works that utilize deep learning for TDVRPs. For example, DDAM-GM [2] designs a dynamic attention mechanism, in which time-dependent travel cost sequences are embedded and selectively weighted through a gated attention unit. SED2AM [3] introduces a temporal locality inductive bias to the encoding module of the policy networks, enabling it to effectively account for the time-dependency in travel distance. While preliminary studies have investigated the application of deep learning to TDVRPs, such approaches typically focus on modeling time-dependent travel costs but do not explicitly explore how to effectively encode asymmetric information. Addressing this limitation will be an important direction for our future research.
>
> In particular, we envision two potential strategies to adapt our model to handle dynamic asymmetries:
>
> 1. Direct Integration of Our Modules into TDVRP Frameworks: Our SVD-based embedding module captures global structural information from asymmetric cost matrices, while the Sinkhorn-based projection layer enforces permutation-invariant constraints that align with route-level consistency. We believe these components can be directly integrated into existing deep learning-based TDVRP models to enhance their representation capacity.
>
> 2. Temporal Tensor Decomposition: Alternatively, we can extend our model to accommodate dynamic asymmetries by introducing a temporal dimension into the cost structure. Specifically, we model the cost matrix as a third-order tensor $\mathbf{C} \in \mathbb{R}^{n \times n \times t}$, where $t$ denotes discrete temporal contexts (e.g., time-of-day intervals or traffic conditions). To obtain time-aware embeddings from such tensor-structured inputs, we may employ low-rank tensor decomposition methods, such as Tucker decomposition [4], or Tensor Train (TT) decomposition [5]. These methods can effectively project $\mathbf{C}$ into a shared low-dimensional latent space.
>
> [1] ICAM:RETHINKING INSTANCE-CONDITIONED ADAPTATION IN NEURAL VEHICLE ROUTING SOLVER
>
> [2] Guo, Feng, et al. "Deep Dynamic Attention Model with Gate Mechanism for Solving Time-dependent Vehicle Routing Problems."
>
> [3] Mozhdehi, Arash, et al. "SED2AM: Solving Multi-Trip Time-Dependent Vehicle Routing Problem using Deep Reinforcement
> Learning." ACM Transactions on Knowledge Discovery from Data 19.5 (2025): 1-33.
>
> [4] Tucker, Ledyard R. "Some mathematical notes on three-mode factor analysis." Psychometrika 31.3 (1966): 279-311.
>
> [5] Oseledets, Ivan V. "Tensor-train decomposition." SIAM Journal on Scientific Computing 33.5 (2011): 2295-2317.

---

> ### Author Response · Authors · 2025-11-26
> **Response (3/4)**
>
> >  **W5: The results of the ACVRPTW problem in Table 3. The method presented in this paper is not the optimal one, but it is highlighted in bold.**
>
> We apologize for the confusion. Our primary goal is to address the limitations of current NCO methods when applied to asymmetric settings, so we emphasize comparisons within learning-based solvers rather than claiming superiority over specialized OR heuristics. The boldface in Table 3 follows the standard NCO convention, indicating the best result among learning-based methods only. To avoid misunderstanding, we have explicitly added this criterion to the footnotes of each table in the revised version.
>
> >  **Q1: It is suggested to conduct comparative experiments on algorithms specifically designed to solve the ACVRP problem(e.g. heuristics and meta-heuristics).**
>
> Thank you for this suggestion-it helps make our empirical evidence more convincing. As discussed in W1, we have made our best effort to reproduce representative ACVRP heuristics and matheuristics.
>
> >  **Q2: The experimental design should incorporate in-distribution scenarios, specifically involving training the model on instances spanning all node sizes and subsequently evaluating its performance on test instances of all sizes.**
>
> Thanks for your suggestions. We talked about mix-size training in W3.
>
> >  **Q3: The results in Table 5 indicate that the existing baseline algorithms are designed for symmetric structure problems and its performance will deteriorate in the case of asymmetry. Therefore, maybe the authors could compare the experimental results of the algorithm proposed in this paper and the baseline algorithm in the symmetric structure scenario?**
>
> We respectfully clarify that the baselines reported in Table 5 are not symmetric-only methods selected for convenience. Rather, they are learning-based solvers that have been explicitly evaluated on asymmetric routing in their original publications and are widely regarded as representative strong baseline under directed/asymmetric settings.
>
> Concretely:
> 1. MatNet and RRNCO are both explicitly proposed to handle asymmetric routing structures.
> 2. UNICO includes, as one of its key contributions, a direct improvement over MatNet and reports results on asymmetric problems.
> 3. ICAM also evaluates on asymmetric tasks and claims improvements over MatNet under the same asymmetric protocols.
>
> Therefore, the degradation observed in Table 5 reflects a substantive limitation that persists even among current asymmetric-capable NCO  solvers, namely their difficulty in preserving and exploiting asymmetric relational representations.

---

> ### Author Response · Authors · 2025-11-26
> **Response (4/4)**
>
> >  **Q4: Regarding the ATSP and ACVRP problems, maybe the authors could add the experimental results of the HGS algorithm?**
>
> For TSP, it is standard to compare against LKH. HGS is designed for CVRP-style problems and includes several VRP-specific components: (i) it encodes solutions as a “giant tour” and uses a split procedure to produce multiple capacity-feasible routes [6]; (ii) its education/local search operates on route-level neighborhoods with load-feasibility tracking and penalty management [7]; and (iii) it employs cross-route operators such as SWAP* that are specific to multi-route settings [8].
>
> We made the following adaptations to enable HGS to handle TSP:
>
> 1. single-vehicle VRP wrapper. For each instance, we randomly select one node as both the starting and ending depot, set the number of vehicles to 1, capacity to 1, and assign zero demand to all nodes.
> 2. Use a TSP-appropriate crossover. We use ordered crossover (OX) which assumes a TSP-style permutation representation.
> 3. Multi-start over depots. We run HGS multiple times per instance; each run uses a newly sampled depot node, and we take the best solution across runs.
>
> For N=100, max runtime is 60s with 6 multi-starts; for N=200, max runtime is 120s with 12 multi-starts; and we continue this linear scaling for larger sizes.
>
> |Method(ATSP)|100 Obj|100 Gap|100 Time|200 Obj|200 Gap|200 Time|500 Obj|500 Gap|500 Time|1000 Obj|1000 Gap|1000 Time|
> |:-|:-:|:-:|:-:|:-:|:-:|:-:|:-:|:-:|:-:|:-:|:-:|:-:|
> |**LKH-100**|1.5643|-|1.08m|1.5721|-|2.44m|1.5763|-|7.32m|1.5739|-|21.92m|
> |**HGS**|1.5779|0.87%|2.09h|1.6708|6.28%|4.19h|1.9053|20.87%|10.74h|2.1356|35.69%|10.00m|
> |**RADAR**|1.5756|0.72%|0.04m|1.5879|1.01%|0.15m|1.6098|2.13%|1.45m|1.6389|4.13%|11.57m|
>
> |Method(ATSP)|In-distributionCost|In-distributionGap(%)|In-distributionTime|OOD(city)Cost|OOD(city)Gap(%)|OOD(city)Time|OOD(cluster)Cost|OOD(cluster)Gap(%)|OOD(cluster)Time|
> |:-|:-:|:-:|:-:|:-:|:-:|:-:|:-:|:-:|:-:|
> |LKH|38.387|*|1.6h|38.903|*|1.6h|12.170|*|1.6h|
> |HGS|38.393|0.02|2.7h|38.909|0.02|2.7h|12.217|0.39|2.7h|
> |MatNet|39.915|3.98|27s|40.548|4.23|27s|12.886|5.88|27s|
> |RRNCO|39.077|1.80|21s|39.783|2.26|21s|12.450|2.30|21s|
> |RADAR|38.671|0.74|21s|39.272|0.95|21s|12.314|1.18|21s|
>
> It can be seen that HGS is not well suited for solving TSP. At size 100, it  reaches comparable solution quality. More importantly, when solving larger sizes, HGS deteriorates sharply within our time budget. We believe that with a longer time budget HGS could improve substantially on these larger instances, but the required runtime would exceed a reasonable range, making it impractical under realistic settings.
>
> For ACVRP, we did not expect HGS to outperform LKH by such a large margin. To make the comparison fair, we ran HGS under two different time budgets aligned with LKH-1000 and LKH-10000. Since HGS can return infeasible solutions, we additionally report the infeasible rate alongside the objective values to provide a complete picture of its practical reliability under each budget.
>
> |Method|100|Gap|Time|200|Gap|Time|500|Gap|Time|1000|Gap|Time|
> |-|-|-|-|-|-|-|-|-|-|-|-|-|
> |HGS-FirstFeasible|5.2245|145.97%|0.11m|5.0882|135.08%|0.36m|4.1051|75.39%|2.03m|5.0098|105.32%|8.96m|
> |HGS-Short|2.1614|1.76%|16.67m|2.0806|-3.88%|25.38m|2.3011|-1.68%|1.08h|2.1094|-13.55%|2.99h|
> |HGS-Long|2.0942|-1.40%|2.78h|1.9733|-8.83%|4.24h|2.1451|-8.35%|10.47h|1.9792|-18.89%|36.33h|
> |LKH-1000|2.1635|1.86%|17.64m|2.1807|0.75%|25.32m|2.3605|0.85%|1.06h|2.4569|0.69%|2.86h|
> |LKH-10000|2.1240|0.00%|2.79h|2.1645|0.00%|4.25h|2.3405|0.00%|10.29h|2.4400|0.00%|36.25h|
> |RADAR|2.1588|1.64%|0.05m|2.1483|-0.75%|0.14m|2.4198|3.39%|1.75m|2.4634|0.96%|11.60m|
>
> |InfeasibleRate|ACVRP100|ACVRP200|ACVRP500|ACVRP1000|
> |:-:|:-:|:-:|:-:|:-:|
> |**HGS-Short**|25.9%|48.7%|92.2%|6.6%|
> |**HGS-Long**|2.5%|0.1%|4.6%|0.0%|
>
> For HGS, we report the average cost over feasible solutions only, since infeasible solutions can occur. We still compute gaps against LKH-10000 as the baseline. The results show that, given a sufficiently large time budget, HGS clearly outperforms both LKH and RADAR in solution quality. However, the key advantage of neural combinatorial optimization is fast, real-time responsiveness: it can produce reasonable solutions within a very short time. To reflect this setting, we also measure the cost of HGS under the first-feasible stopping rule (HGS-FirstFeasible). Under a time budget comparable to NCO methods such as RADAR, HGS performs very poorly: the obtained solutions are far from optimal, and the limited time is insufficient for HGS to refine them toward high-quality or near-optimal routes. This highlights that HGS relies on long runtime to realize its strength, whereas NCO methods retain strong performance in the low-latency regime.
>
> [6] Where to Split in Hybrid Genetic Search for the Capacitated Vehicle Routing Problem
>
> [7] A GRASP Hybrid Genetic Algorithm for the Capacitated Vehicle Routing Problem
>
> [8] https://github.com/vidalt/HGS-CVRP

---

> ### Comment · Reviewer_D1Sv · 2025-11-26
>
> The author answered most of my questions and I will raise my score.

---

> > ### Author Response · Authors · 2025-11-27
> >
> > Thank you for acknowledging our rebuttal and for raising the score. If you have any further questions or concerns, please feel free to let us know, and we are more than happy to clarify anything before the discussion period closes in the coming days.

---

### Official Review · Reviewer_24SW · 2025-10-31

**Soundness:** 3
**Presentation:** 2
**Contribution:** 2
**Rating:** 6
**Confidence:** 3

**Summary:**

The paper proposes RADAR, a neural framework for solving Vehicle Routing Problems (VRPs) with asymmetric distance matrices. RADAR addresses both static and dynamic asymmetries via two key components that learn asymmetry-aware embeddings. For static asymmetry, it applies Singular Value Decomposition (SVD) to the input distance matrix to obtain asymmetry-aware node embeddings. For dynamic asymmetry, i.e., asymmetries that emerge during encoder attention, RADAR replaces the standard softmax with Sinkhorn normalization within the attention mechanism.

**Strengths:**

1.	The SVD-based embedding is simple yet empirically effective. The motivation is clearly articulated and technically convincing.

2.	The paper presents extensive experiments across multiple datasets and VRP variants, providing solid empirical support for the method’s effectiveness.

**Weaknesses:**

While leveraging SVD to encode matrix asymmetry is a valuable idea, the current contribution is confined to routing problems. Asymmetric (Directed) edge structures widely exist in other graph-based combinatorial optimization tasks. This scope limitation reduces the broader impact and significance of the work. If the authors could extend the idea from routing distance asymmetry to general graph asymmetry and validate it on a more diverse set of problems, the contribution would be substantially strengthened.

**Questions:**

The proposed SVD-based embedding appears tailored to attention-based models. Although attention mechanisms are mainstream in many domains, it remains important to assess effectiveness under alternative network architectures. Such an analysis would clarify whether the observed gains are specific to certain architectures or are more generally applicable. Could the authors provide a discussion and, if possible, experiments on the choice of network architecture?

---

> ### Author Response · Authors · 2025-11-26
> **Response (1/2)**
>
> >  **W: While leveraging SVD to encode matrix asymmetry is a valuable idea, the current contribution is confined to routing problems. Asymmetric (Directed) edge structures widely exist in other graph-based combinatorial optimization tasks. This scope limitation reduces the broader impact and significance of the work. If the authors could extend the idea from routing distance asymmetry to general graph asymmetry and validate it on a more diverse set of problems, the contribution would be substantially strengthened.**
>
> Thank you for this insightful comment. We agree the current experiments focus on routing, but the core idea is not specific to “distance matrices.” Our method targets a broader class of relational features : any task where the problem instance is naturally represented as a relation matrix R (not necessarily metric distances). By decomposing R into asymmetric components as initialization and injecting them into attention, the model can recover and propagate relational signals during message aggregation. In this sense, once a problem provides a relational matrix (costs, compatibilities, precedences, processing relations, transition penalties, etc.), the same decomposition-plus-attention mechanism applies directly.
>
> A concrete example is Flexible Flow Shop Scheduling Problem (FFSP). FFSP involves multiple production stages, each with several parallel machines; each job must be processed stage by stage, and the decision is to assign jobs to machines and order them to minimize makespan or total completion time. In MatNet, FFSP instances are encoded as matrices of relations, processing-time matrices between jobs and machines at each stage. These entries are relational features rather than geometric distances: they describe directed interactions like job i processed on machine m time.
>
> To further demonstrate this generality beyond routing, we conducted an additional experiment on MatNet  under the FFSP setting. Specifically, we replaced MatNet’s original initialization with our asymmetric SVD-based relational-feature encoding, and retrained the resulting model on size N=20. We then evaluated zero-shot generalization to larger OOD sizes N=50 and N=100.
>
> |Method|20|Gap|Time|50|Gap|Time|100|Gap|Time|
> |-|-|-|-|-|-|-|-|-|-|
> |CPLEX(60s)|46.4|65.9|17h|×|×|×|×|×|×|
> |CPLEX(600s)|36.6|30.8|167h|×|×|×|×|×|×|
> |Random|47.8|70.9|1m|93.2|78.2|2m|167.2|80.9|3m|
> |ShortestJobFirst|31.3|11.9|40s|57.0|9.0|1m|99.3|7.4|2m|
> |GeneticAlgorithm|30.6|9.4|7h|56.4|7.8|16h|98.7|6.8|29h|
> |ParticleSwarmOpt.|29.1|4.0|13h|55.1|5.3|26h|97.3|5.2|48h|
> |MatNet|28.0676|0.3|0.82m|52.5973|0.6|1.36m|93.0295|0.6|2.39m|
> |MatNet\_SVD|**27.9727**|0.0|0.85m|**52.3042**|0.0|1.39m|**92.4473**|0.0|2.43m|
>
> We report MS (makespan) as the primary objective, which measures the completion time of the last job in the schedule; lower MS indicates a better schedule with shorter overall production horizon. Under this greedy inference setting, replacing MatNet’s original matrix encoder with our SVD-based relational-feature encoding yields consistent improvements across sizes.

---

> ### Author Response · Authors · 2025-11-26
> **Response (2/2)**
>
> >  **Q: The proposed SVD-based embedding appears tailored to attention-based models. Although attention mechanisms are mainstream in many domains, it remains important to assess effectiveness under alternative network architectures. Such an analysis would clarify whether the observed gains are specific to certain architectures or are more generally applicable. Could the authors provide a discussion and, if possible, experiments on the choice of network architecture?**
>
> We agree with the reviewer that our SVD-based embedding is tailored to attention-based models. Concretely, our decomposition is designed to exploit the dot-product attention structure: the asymmetric relational signal is factorized so that it can be reconstructed inside the Q·K interaction and thus directly influence attention scores. So the main theoretical guarantee is inherently tied to architectures that compute similarity via dot products.
>
> We did look for public non-attention alternatives on asymmetric routing. To our knowledge, the only clearly non-attention direction that has been explored for routing is the GCN-style heatmap prediction line, which avoids autoregressive attention decoding. In our own attempts, the original AFGN [1] and DeepACO [2] did not converge on asymmetric tasks without substantial redesign. At present, only GREAT [3] reports results on asymmetric tasks, but its code has not been released, so we cannot fairly benchmark against it.
>
> So, we are happy to run additional experiments under non-attention architectures if a suitable public asymmetric-routing baseline is available. If the reviewer knows of any open-source, genuinely non-attention framework that reliably handles ATSP/ACVRP-type asymmetry, we would be very glad to test our embedding there and add the results in the revision.
>
> [1] AFGN: Adaptive Filtering Graph Neural Network for Few-Shot Learning
>
> [2] DeepACO: Neural-enhanced Ant Systems for Combinatorial Optimization
>
> [3] A GREAT Architecture for Edge-Based Graph Problems Like TSP

---

> > ### Comment · Reviewer_24SW · 2025-11-27
> >
> > Thank you for your response. The additional experiments and discussions have adequately addressed my concerns. I hope that the versatility of the SVD-based attention mechanism will be clearly demonstrated in the final version of the paper. I am raising my score to an accept.

---

> > > ### Author Response · Authors · 2025-11-27
> > >
> > > Thank you very much for the positive assessment and for raising your score. We truly appreciate your recognition of our work. We have updated the Appendix I to include results on scheduling tasks, which can further illustrate the applicability of our SVD-based relational-feature initialization beyond routing. We value your recommendation, and in future revisions or follow-up studies we plan to incorporate more experiments to more fully demonstrate the versatility of the relational-feature mechanism.

---

### Official Review · Reviewer_c4fR · 2025-10-31

**Soundness:** 3
**Presentation:** 3
**Contribution:** 2
**Rating:** 4
**Confidence:** 4

**Summary:**

The paper addresses the problem of learning to solve vehicle routing problem,
esp. with asymmetric distances. The authors propose a method with two
aspects:
- i) to concatenate the left and right eigenvectors of the SVD of the distance
  matrix as customer features (replacing customer coordinates of the symmetric
  case), and
- ii) to make the attention matrix doubly stochastic with a Sinkhorn transform
  (instead of making it row stochastic by a softmax).

In experiments on synthetic and real TSP and CVRP problems they show
that their method outperforms existing neural routing methods, esp.
for larger out-of-distribution instances, e.g., for ACVRP100 only by 0.3%,
but for ACVRP1000 by 38%

**Strengths:**

- s1. two simple ideas: using left and right eigenvectors for asymmetric distances and making attention
  doubly stochastic.
- s2. many experiments, also some on different problem variations.
- s3. well written.

**Weaknesses:**

- w1. ablation study and thus the attribution of improvement to the two ideas is not fully clear.
- w2. performance of OR heuristics is not fully clear.
- w3. why the performance improvements are so much larger for larger problems or
  out of distribution problems is not investigated.
- w4. small methodological contribution: adding the eigenvectors is basically feature engineering the asymmetric
  distance matrix characteristics of the problem.

more details:

w1. ablation study and thus the attribution of improvement to the two ideas is not fully clear.
- You could add eigenvector features basically to any method with customer features.
- And you could use the Sinkhorn normalization with any method involving on attention.
- Which of the two contributions,
  - i) using eigenvectors of the distance matrix as customer features and
  - ii) making the attention matrix douvly stochastic
  is contributing how much to the success of the method?
- The example you provide in fig. 5 is problematic:
  - You compare on ATSP100, but on ATSP100 lifts of your method are very small.
    It might be more convincing to also compare on ACVRP1000 where you have substantial lifts.
  - You report for your method RADAR w/o sinkhorn an objective value of ~1.5925,
    but in tab. 1 your closest baseline ReLD+ achieves an slightly even better objective:
	1.5900. So in this case, adding eigenvectors seems not to help?

w2. performance of OR heuristics is not fully clear.
- Why is the OR solver HGS not compared in tab. 1 and 3?
- Why do we not see runtimes in tab. 2? W/o. runtimes comparing VRP solvers is not so meaningful.

w3. why the performance improvements are so much larger for larger problems or
out of distribution problems is not investigated.
- This is an odd phenomenon: on the problems you train for, ACVRP100, you see only
  a very small lift of 0.3%, but then on the larger out-of-distribution instances, e.g.,
  for ACVRP1000 the lift is a substantial 38%. Which aspect of your method
  is responsible for this behavior?

w4. small methodological contribution:
- adding the eigenvectors is basically feature engineering the asymmetric
  distance matrix characteristics of the problem.

Small points:
- p1. It would be helpful to show in tab. 1 which of the baselines have access to which problem
  information such as the asymmetric distance matrix, the demands etc.

**Questions:**

- q1. Can you separate the effects of your two contributions i) and ii) clearly in an ablation study?
  Which of the baselines can be equipped with features i) or doubly stochastic attention scores ii)
  and how do they perform then?
- q2. How does HGS perform on problems in tab. 1 and 3? Do you really observe
  a lift over OR heuristics?
- q3. Your method performs better esp. in larger out-of-distribution settings. Can you explain or
  provide some first analysis why this happens?

---

> ### Author Response · Authors · 2025-11-26
> **Response (1/6)**
>
> >  **W1: ablation study and thus the attribution of improvement to the two ideas is not fully clear.**
>
> >  **1. You could add eigenvector features basically to any method with customer features. And you could use the Sinkhorn normalization with any method involving on attention.**
>
> We agree with the reviewer that, in principle, eigenvector-based features and Sinkhorn normalization could be incorporated into many existing neural VRP solvers with attention layers. This generality is exactly what we aim for with RADAR. Our claims are twofold:
>
> (1) models that originally only consume coordinates can be made capable of handling asymmetric costs through our representation,
>
> (2) current solvers often fail to learn sufficiently rich representations of asymmetry, whereas adding our modules significantly alleviates this issue.
>
> To support (1), Section 4.2 shows that extending Routefinder with RADAR enables it to solve all 16 asymmetric settings we consider. To support (2), we integrate RADAR into RR-NCO and observe clear gains on real-world asymmetric benchmarks.
>
> Our goal is therefore not to argue that only our architecture can use SVD or Sinkhorn, but to provide a principled, empirically validated recipe for making attention-based NCO models robust to asymmetry and to encourage the community to treat asymmetric VRPs as a first-class target in future work.
>
> >  **2. Which of the two contributions,
> i) using eigenvectors of the distance matrix as customer features and
> ii) making the attention matrix doubly stochastic is contributing how much to the success of the method?**
>
> Thank you for this insightful question. Our additional 2×2 ablation (with/without SVD × with/without Sinkhorn) was designed precisely to disentangle these two contributions. From these results, we see three consistent patterns:
>
> - First, adding SVD alone already improves over the plain model across all sizes.
> - Second, enabling Sinkhorn alone can get a better performance than SVD alone.
> - Third, combining **SVD + Sinkhorn** gives the best results overall, showing that SVD still provides a meaningful, complementary benefit on top of Sinkhorn by preserving more global structure of the distance matrix in the node features.
>
> |Method|SVD|Sink|100|Gap|Time|200|Gap|Time|500|Gap|Time|1000|Gap|Time|
> |-|-|-|-|-|-|-|-|-|-|-|-|-|-|-|
> |RADAR|✗|✗|1.5969|2.08|0.02m|1.6543|5.23|0.13m|1.8610|18.06|1.34m|2.1821|38.64|11.11m|
> |RADAR|✓|✗|1.5928|1.82|0.03m|1.6273|3.51|0.14m|1.7418|10.50|1.44m|1.9342|22.89|11.45m|
> |RADAR|✗|✓|1.5829|1.19|0.03m|1.6007|1.82|0.13m|1.6379|3.91|1.43m|1.6878|7.24|11.37m|
> |RADAR|✓|✓|**1.5756**|**0.72**|0.04m|**1.5879**|**1.01**|0.15m|**1.6098**|**2.13**|1.45m|**1.6389**|**4.13**|11.57m|
>
> In summary, we view eigenvector features and Sinkhorn attention not as competing mechanisms, but as complementary: SVD focuses on encoding static asymmetry in the inputs, and Sinkhorn focuses on enforcing a better-behaved attention geometry during inference.

---

> ### Author Response · Authors · 2025-11-26
> **Response (2/6)**
>
> >  **3. The example you provide in fig. 5 is problematic:
> (1) You compare on ATSP100, but on ATSP100 lifts of your method are very small.**
>
> We appreciate the reviewer’s comment about Fig. 5. Our intention there was to illustrate the training dynamics of Sinkhorn versus softmax, so we plotted curves on ATSP100, which is the actual training size used in our main experiments. We do not train on ACVRP1000, since training directly at N = 1000 would require prohibitive computational resources. The goal of Fig. 5 is therefore not to showcase the largest absolute lift, but to demonstrate that even on ATSP100, where the final performance gap is relatively modest, replacing softmax with Sinkhorn already yields faster convergence, more stable training, and consistently better final objective values.
>
> >  **3. (2) You report for your method RADAR w/o sinkhorn an objective value of ~1.5925, but in tab. 1 your closest baseline ReLD+ achieves an slightly even better objective: 1.5900. So in this case, adding eigenvectors seems not to help?**
>
> Thank you for raising this point. First, we would like to clarify that these two numbers are not directly comparable: they are obtained on **different datasets**. The ~1.5925 value comes from the training curve on the ATSP100 *training* distribution, whereas 1.5900 is the average objective on a separate *test* set. Since the train and test sets need not have the same difficulty, their absolute numbers cannot be meaningfully compared to judge whether SVD helps or not.
>
> Second, If you are interested in a direct comparison between “our method with only SVD” and ReLD, we are happy to provide these results. Our SVD-only variant (RADAR_SVD) can indeed be slightly weaker than ReLD on the ATSP100 test set, but it consistently exhibits better generalization on larger instances. However, this is not a fair way to assess the effect of SVD, because ReLD includes other architectural improvements.
>
> |Method|100|GAP|Time|200|GAP|Time|500|GAP|Time|1000|GAP|Time|
> |:-|:-:|:-:|:-:|:-:|:-:|:-:|:-:|:-:|:-:|:-:|:-:|:-:|
> |ReLD|**1.5900**|**1.64%**|0.03m|1.6310|3.75%|0.15m|1.7873|13.39%|1.49m|2.0723|31.67%|11.11m|
> |RADAR_SVD|1.5928|1.82%|0.03m|**1.6273**|**3.51%**|0.17m|**1.7418**|**10.50%**|1.44m|**1.9342**|**22.89%**|11.45m|
>
> Third, if the goal is to isolate the effect of SVD, a more appropriate comparison is within the same backbone. In the rebuttal, we therefore report: (i) RADAR without SVD versus RADAR with SVD, and (ii) ReLD versus ReLD augmented with SVD-based initialization. In both cases, adding SVD yields consistent improvements.
>
> |Method|SVD|100|GAP|Time|200|GAP|Time|500|GAP|Time|1000|GAP|Time|
> |:-|:-:|:-:|:-:|:-:|:-:|:-:|:-:|:-:|:-:|:-:|:-:|:-:|:-:|
> |ReLD|✗|1.5900|1.64%|0.03m|1.6310|3.75%|0.15m|1.7873|13.39%|1.49m|2.0723|31.67%|11.11m|
> |ReLD|✓|**1.5824**|**1.16%**|0.03m|**1.6128**|**2.59%**|0.17m|**1.7019**|**7.97%**|1.60m|**1.8398**|**16.89%**|12.42m|
>
> |Method|SVD|100|GAP|Time|200|GAP|Time|500|GAP|Time|1000|GAP|Time|
> |:-|:-:|:-:|:-:|:-:|:-:|:-:|:-:|:-:|:-:|:-:|:-:|:-:|:-:|
> |RADAR|✗|1.5969|2.08%|0.02m|1.6543|5.23%|0.13m|1.8610|18.06%|1.34m|2.1821|38.64%|11.11m|
> |RADAR|✓|**1.5928**|**1.82%**|0.03m|**1.6273**|**3.51%**|0.17m|**1.7418**|**10.50%**|1.44m|**1.9342**|**22.89%**|11.45m|
>
> >  **W2: performance of OR heuristics is not fully clear.**
>
> >  **1. Why is the OR solver HGS not compared in tab. 1 and 3?**
>
> Thank you for pointing this out. Our evaluation protocol follows the three standard settings used in MatNet, RouteFinder, and RRNCO so that comparisons remain directly aligned with prior work. We apologize for not reporting HGS in synthetic ACVRP setting; this was an oversight, because we expected LKH to be a sufficient classical reference there.
>
> For TSP, it is standard to compare with LKH, since LKH is a strong reference solver in the single-tour regime. In contrast, HGS is designed for CVRP-style problems: (i) it encodes solutions as a “giant tour” and uses a split procedure to produce multiple capacity-feasible routes [1]; (ii) its local search operates on route-level neighborhoods with load-feasibility tracking and penalty management [2]; and (iii) it employs cross-route operators such as SWAP* that are specific to multi-route settings [3].  Applying HGS to TSP would therefore require non-trivial modifications and adaptations, and we are happy to describe these in detail and report corresponding results in Q2.
>
> >  **2. Why do we not see runtimes in tab. 2? W/o. runtimes comparing VRP solvers is not so meaningful.**
>
> We agree that runtimes are important for a meaningful comparison of VRP solvers. For the 16 asymmetric settings in Table 2, the full set of results is quite large, so we report them in the appendix as Table 8 rather than in the main text due to space constraints.
>
> [1] Where to Split in Hybrid Genetic Search for the Capacitated Vehicle Routing Problem
>
> [2] A GRASP Hybrid Genetic Algorithm for the Capacitated Vehicle Routing Problem
>
> [3] https://github.com/vidalt/HGS-CVRP

---

> ### Author Response · Authors · 2025-11-26
> **Response (3/6)**
>
> >  **W3: why the performance improvements are so much larger for larger problems or out of distribution problems is not investigated. This is an odd phenomenon: on the problems you train for, ACVRP100, you see only a very small lift of 0.3%, but then on the larger out-of-distribution instances, e.g., for ACVRP1000 the lift is a substantial 38%. Which aspect of your method is responsible for this behavior?**
>
> We agree that the gap reduction on ACVRP100 appears numerically small compared to the much larger gains on ACVRP1000. However, these two regimes are not directly comparable. A large body of machine learning literature shows that standard empirical risk minimization models often achieve strong ID accuracy. A well-recognized reason is that ERM models often latch onto spurious correlations [4]/ shortcut features [5] that are easy to fit and highly predictive only within the training domain; such “shortcuts” can yield strong ID results quickly but fail under domain shift. Therefore, ID benchmarks often become statistically saturated. In such case, additional improvements targeted at robustness or invariance may translate into only small absolute gains on ID, while producing much larger improvements under distribution shift, where shortcut-reliant baselines collapse. This is a standard and widely observed pattern in distribution-shift/domain-generalization research.
>
> We will elaborate why our methods performs well in out-of-distribution datasets.
>
> The current fixed top-$k$ initialization introduces a scale-dependent distribution shift. By keeping only the top-$k$ distances per node, they effectively project the original $O(n\times n)$ cost matrix to a sparse $O(nk)$ graph, so the retained information ratio is approximately
>
> $$
> \frac{nk}{n(n-1)} \approx \frac{k}{n}.
> $$
>
> This means the initialization preserves about $10\%$ of distance information at $n=100$ (losing $\sim 90\%$), but only about $1\%$ at $n=1000$ (losing $\sim 99\%$). Although the rule “top-10 neighbors” is unchanged, the statistics of the initialized input change systematically with $n$: relative graph density, neighborhood coverage, and the visibility of non-local costs all shrink as $1/n$. Consequently, a policy trained on ACVRP100 implicitly learns under a regime where roughly $10\%$ of structural information is available, but at ACVRP1000 it must operate with only $\sim 1\%$, so the semantic meaning of the initialized features is no longer comparable.
>
> Using the top-$k$ singular values from SVD, we define the information ratio as
>
> $$
> \rho_k = \frac{\sum_{i=1}^{k}\sigma_i^2}{\sum_{i=1}^{n}\sigma_i^2}
> = \frac{\sum_{i=1}^{k}\sigma_i^2}{\lVert A\rVert_F^2},
> $$
>
> where $\sigma_i$ is the $i$-th singular value and $\lVert A\rVert_F$ denotes the Frobenius norm of the cost matrix. In our experiments with $k=10$, $\rho_{10}$ decreases only moderately from **84.44%** on ATSP100 to **74.45%** on ATSP1000. This shows that, even with a fixed number of spectral components, SVD retains most of the matrix energy across scales: the global structure is largely preserved, and the amount of information discarded does not grow dramatically as $n$ increases.
>
> Sinkhorn normalization improves generalization because it introduces an explicit column-communication mechanism on top of the usual row-wise Softmax. Since Markdown cannot render the visualization, we provide the corresponding two-panel attention maps in Figure 8 of Appendix G. With standard Softmax attention (Fig. 8(a)), each row is normalized independently, so the model can assign disproportionately large probability mass to a few “attractor” nodes that naturally receive high incoming attention from many rows. This creates strong column imbalance: a small set of nodes dominate the attention matrix, while most columns stay uniformly low. During decoding, once those high-attention nodes are visited and removed from the candidate set, the remaining columns are all similarly small, so the attention distribution becomes nearly flat and the model is forced to make decisions under high uncertainty because there is no clear preference signal left. Sinkhorn (Fig. 8(b)) iteratively normalizes both rows and columns. This prevents any single column from absorbing too much global mass and redistributes attention more evenly across nodes. As a result, even after some nodes are selected, the remaining nodes still maintain meaningful relative attention differences instead of collapsing to a uniform low baseline. In practice, this makes the decoder’s selection policy more stable and less dependent on distribution-specific “hub” patterns.
>
> [4] In Search of Lost Domain Generalization
>
> [5] Shortcut Learning in Deep Neural Networks

---

> ### Author Response · Authors · 2025-11-26
> **Response (4/6)**
>
> >  **W4: small methodological contribution: adding the eigenvectors is basically feature engineering the asymmetric distance matrix characteristics of the problem.**
>
> We respectfully disagree that the methodological contribution is “small” merely because the mechanism can be described as adding eigenvector features. In our view, the significance should be judged relative to the needs of the NCO community rather than the algorithmic complexity of the building blocks. Our literature review suggests that, for asymmetric VRPs, representation is indeed a bottleneck: existing designs such as the dual embedding streams in MatNet or the top-k neighbor encodings in RRNCO and related methods all introduce non-trivial limitations.
>
> Our SVD-based initialization is not an ad-hoc “feature engineering” trick chosen purely for empirical gains, but a well-motivated representation derived from our static asymmetry analysis. It is explicitly constructed to approximately reconstruct the original distance matrix from the node embeddings, so that the model sees a compact yet theoretically grounded encoding of global asymmetric structure rather than a heavily truncated local view. While the mechanism itself is simple, we believe the formulation, analysis, and empirical validation together provide useful insight for the community and offer a principled way to rethink initialization for asymmetric NCO.
>
> >  **Small points: p1. It would be helpful to show in tab. 1 which of the baselines have access to which problem information such as the asymmetric distance matrix, the demands etc.**
>
> We thank the reviewer for this suggestion. In each table, all compared methods use the same problem features: for Table 1, all ATSP models take the asymmetric distance matrix as input, and all ACVRP models take the asymmetric distance matrix plus demands as input. No method has access to additional information.
>
> >  **Q1. Can you separate the effects of your two contributions i) and ii) clearly in an ablation study? Which of the baselines can be equipped with features i) or doubly stochastic attention scores ii) and how do they perform then?**
>
> In W1.2, we provide a full 2×2 study (with/without SVD × with/without Sinkhorn), which cleanly separates the effects of contribution (i) and (ii). The results show that both SVD and Sinkhorn improve generalization, with Sinkhorn providing the largest individual gain, while the combination of SVD + Sinkhorn consistently achieves the best performance across scales.
>
> Regarding which baselines can be equipped with these components: any attention-based neural VRP solver can use our methods. Our goal is to propose a *plug-in* representation/attention module, not a new standalone architecture. To make this concrete, we equip ReLD with our methods; in both cases, ReLD’s performance improves, especially on larger and more asymmetric instances. This supports our claim that SVD and Sinkhorn are generic, modular enhancements that can strengthen existing attention-based NCO models.
>
> |Method|SVD|Sink|100|Gap|Time|200|Gap|Time|500|Gap|Time|1000|Gap|Time|
> |-|-|-|-|-|-|-|-|-|-|-|-|-|-|-|
> |ReLD|✗|✗|1.5900|1.64%|0.03m|1.6310|3.75%|0.15m|1.7873|13.39%|1.49m|2.0723|31.67%|12.18m|
> |ReLD|✓|✗|1.5842|1.27%|0.04m|1.6128|2.59%|0.17m|1.7019|7.79%|1.60m|1.8398|16.69%|12.42m|
> |ReLD|✓|✓|**1.5745**|**0.65%**|0.05m|**1.5873**|**0.97%**|0.18m|**1.6091**|**2.08%**|1.67m|**1.6329**|**3.56%**|12.56m|

---

> ### Author Response · Authors · 2025-11-26
> **Response (5/6)**
>
> >  **Q2. How does HGS perform on problems in tab. 1 and 3? Do you really observe a lift over OR heuristics?**
>
> To apply HGS within PyVRP to TSP, we made the following straightforward adaptations:
> 1. single-vehicle VRP wrapper. For each instance, we randomly select one node as both the starting and ending depot, set the number of vehicles to 1, capacity to 1, and assign zero demand to all nodes. This converts TSP into a special CVRP case that PyVRP can parse, since PyVRP’s HGS interface requires an explicit depot, demand vector, and capacity parameter.
> 2. Use a TSP-appropriate crossover. We replace PyVRP’s default VRP crossover with ordered crossover (OX). The PyVRP documentation explicitly notes that OX assumes a TSP-style permutation representation.
> 3. Multi-start over depots. We run HGS multiple times per instance; each run uses a newly sampled depot node, and we take the best solution across runs. This is equivalent to a multi-start heuristic for TSP.
>
> For N=100, max runtime is 60s with 6 multi-starts; for N=200, max runtime is 120s with 12 multi-starts; and we continue this linear scaling for larger sizes.
>
> |Method(ATSP)|100 Obj|100 Gap|100 Time|200 Obj|200 Gap|200 Time|500 Obj|500 Gap|500 Time|1000 Obj|1000 Gap|1000 Time|
> |:-|:-:|:-:|:-:|:-:|:-:|:-:|:-:|:-:|:-:|:-:|:-:|:-:|
> |**LKH-100**|1.5643|-|1.08m|1.5721|-|2.44m|1.5763|-|7.32m|1.5739|-|21.92m|
> |**HGS**|1.5779|0.87%|2.09h|1.6708|6.28%|4.19h|1.9053|20.87%|10.74h|2.1356|35.69%|10.00m|
> |**RADAR**|1.5756|0.72%|0.04m|1.5879|1.01%|0.15m|1.6098|2.13%|1.45m|1.6389|4.13%|11.57m|
>
> |Method(ATSP)|In-distributionCost|In-distributionGap(%)|In-distributionTime|OOD(city)Cost|OOD(city)Gap(%)|OOD(city)Time|OOD(cluster)Cost|OOD(cluster)Gap(%)|OOD(cluster)Time|
> |:-|:-:|:-:|:-:|:-:|:-:|:-:|:-:|:-:|:-:|
> |LKH|38.387|*|1.6h|38.903|*|1.6h|12.170|*|1.6h|
> |HGS|38.393|0.02|2.7h|38.909|0.02|2.7h|12.217|0.39|2.7h|
> |MatNet|39.915|3.98|27s|40.548|4.23|27s|12.886|5.88|27s|
> |RRNCO|39.077|1.80|21s|39.783|2.26|21s|12.450|2.30|21s|
> |RADAR|38.671|0.74|21s|39.272|0.95|21s|12.314|1.18|21s|
>
> Moreover, in the first table, HGS runs over 100× longer than LKH, whereas in the second table HGS takes less than 2× the LKH runtime. As we noted in the paper, even when LKH is given substantially more time, its cost does not decrease markedly; therefore we report LKH under the smallest reasonable time budget for a fair efficiency comparison. The runtimes in the second table are taken directly from the RRNCO. It can be seen that HGS is not well suited for solving TSP. At size 100, it reaches comparable solution quality. More importantly, when solving larger sizes, HGS deteriorates sharply within our time budget. We believe that with a longer time budget HGS could improve substantially on these larger instances, but the required runtime would exceed a reasonable range, making it impractical under realistic settings.
>
> For ACVRP, we did not expect HGS to outperform LKH by such a large margin. To make the comparison fair, we ran HGS under two different time budgets aligned with LKH-1000 and LKH-10000. Since HGS can return infeasible solutions, we additionally report the infeasible rate alongside the objective values to provide a complete picture of its practical reliability under each budget.
>
> |Method|100|Gap|Time|200|Gap|Time|500|Gap|Time|1000|Gap|Time|
> |-|-|-|-|-|-|-|-|-|-|-|-|-|
> |HGS-FirstFeasible|5.2245|145.97%|0.11m|5.0882|135.08%|0.36m|4.1051|75.39%|2.03m|5.0098|105.32%|8.96m|
> |HGS-Short|2.1614|1.76%|16.67m|2.0806|-3.88%|25.38m|2.3011|-1.68%|1.08h|2.1094|-13.55%|2.99h|
> |HGS-Long|2.0942|-1.40%|2.78h|1.9733|-8.83%|4.24h|2.1451|-8.35%|10.47h|1.9792|-18.89%|36.33h|
> |LKH-1000|2.1635|1.86%|17.64m|2.1807|0.75%|25.32m|2.3605|0.85%|1.06h|2.4569|0.69%|2.86h|
> |LKH-10000|2.1240|0.00%|2.79h|2.1645|0.00%|4.25h|2.3405|0.00%|10.29h|2.4400|0.00%|36.25h|
> |RADAR|2.1588|1.64%|0.05m|2.1483|-0.75%|0.14m|2.4198|3.39%|1.75m|2.4634|0.96%|11.60m|
>
> |InfeasibleRate|ACVRP100|ACVRP200|ACVRP500|ACVRP1000|
> |:-:|:-:|:-:|:-:|:-:|
> |**HGS-Short**|25.9%|48.7%|92.2%|6.6%|
> |**HGS-Long**|2.5%|0.1%|4.6%|0.0%|
>
> For HGS, we report the average cost over feasible solutions, since infeasible solutions can occur. We still compute gaps against LKH-10000 as the baseline. The results show that, given a sufficiently large time budget, HGS clearly outperforms both LKH and RADAR in solution quality. However, the key advantage of neural combinatorial optimization is fast, real-time responsiveness: it can produce reasonable solutions within a very short time. To reflect this setting, we also measure the cost of HGS under the first-feasible stopping rule (HGS-FirstFeasible). Under a time budget comparable to NCO methods such as RADAR, HGS performs very poorly: the obtained solutions are far from optimal, and the limited time is insufficient for HGS to refine them toward high-quality or near-optimal routes. This highlights that HGS relies on long runtime to realize its strength, whereas NCO methods retain strong performance in the low-latency regime.

---

> ### Author Response · Authors · 2025-11-26
> **Response (6/6)**
>
> >  **Q3. Your method performs better esp. in larger out-of-distribution settings. Can you explain or provide some first analysis why this happens?**
>
> Thank you for this question. We have already provided a detailed explanation in our W3 response, and here we add more findings on two components we proposed. Following the protocol in the original paper, we further conduct experiments across different asymmetry levels a. The results show a clear trend: as
> a increases and the problem becomes more strongly asymmetric, the performance gain from Sinkhorn normalization and SVD becomes larger.
>
> |Method|SVD|Sinkhorn|Low|GAP|Medium|GAP|High|GAP|
> |-|-|-|-|-|-|-|-|-|
> |RADAR|✗|✗|4.7010|0.00%|4.4867|0.00%|4.0342|0.00%|
> |RADAR|✓|✗|4.6198|-1.73%|4.3933|-2.08%|3.9268|-2.66%|
> |RADAR|✓|✓|4.6115|-1.90%|4.3743|-2.51%|3.8784|-3.86%|

---

> > ### Comment · Reviewer_c4fR · 2025-11-27
> >
> > Thanks to the authors for their detailed answers with several new empirical
> > evaluations. My questions have been answered and my concerns addressed,
> > I updated my score accordingly.

---

> > > ### Author Response · Authors · 2025-11-27
> > >
> > > Thank you very much for raising the score and taking the time to review our work so carefully. We appreciate your thoughtful questions and are glad that the additional experiments helped address your concerns. Your support and score update mean a lot to us.

---

### Official Review · Reviewer_qwgM · 2025-10-31

**Soundness:** 2
**Presentation:** 3
**Contribution:** 2
**Rating:** 4
**Confidence:** 4

**Summary:**

This paper addresses the challenge of solving asymmetric vehicle routing problems (AVRPs) with neural solvers. The authors propose RADAR, a framework that modifies existing neural solvers to handle asymmetric distance matrices. The contribution is twofold: (1) an SVD-based initialization to encode static asymmetry into compact node embedding, and (2) the use of Sinkhorn normalization in the attention mechanism. The authors validate RADAR on a wide range of synthetic and real-world asymmetric VRPs, showing superior performance and generalization compared to baselines.

**Strengths:**

1. The paper is well-written, and the methodology is explained clearly.

2. The conceptual framing of the problem into "static" and "dynamic" asymmetry is interesting, providing a new perspective for addressing asymmetric routing problems.

3. The experimental evaluation is a major strength. It is comprehensive, rigorous, and includes a wide array of benchmarks

4. The empirical evaluation showcases the strong performance of the RADAR compared to the SOTA baseline, especially in out-of-distribution generalization to larger problem sizes.

**Weaknesses:**

1. The paper's contribution lies in the clever application of existing techniques rather than the development of fundamentally new methods. Both SVD and Sinkhorn normalization are standard, well-established algorithms. While the engineering is solid, the work feels more incremental than transformative from a methodological standpoint.

2. In Table 1, the authors state that all retrained baselines were evaluated using z-score normalization. However, the authors do not justify why this specific normalization was uniformly applied. Applying a non-native normalization may unfairly penalize baseline performance, making it unclear if the comparison was conducted under optimal conditions for all methods.

3. The paper uses different normalizations for different datasets. However, the paper provides no ablation study to compare the effect of z-score versus min-max (or no normalization) on the same dataset. It remains unclear whether the impressive performance gains are fully attributable to the proposed SVD and Sinkhorn modules or are partially an artifact of a specific, un-analyzed normalization choice.

4.  The ablation study in Table 6 severely weakens the case for the SVD-based initialization as a standalone contribution. The RADAR-SVD performs very poorly on larger-scale problems, achieving a gap of over 20% on ATSP1000. This poor generalization suggests that the strong performance is critically dependent on the Sinkhorn operation.

5. Some critical hyperparameters' details are missing. For example, the number of iterations T for Sinkhorn normalization is not specified. It is unclear whether the increasing number of iterations significantly increases the computational load and performance of the model.

**Questions:**

1. To verify the necessity of the proposed SVD initialization, could you provide results for two models that keep all other settings unchanged (including z-score normalization) and combine Sinkhorn normalization with forward-only distance features (i.e., top-10 nearest outgoing neighbors) and bidirectional distance features (i.e., concatenating top-10 outgoing and top-10 incoming neighbor distances), respectively?

2. What is the number of iterations T used for Sinkhorn normalization?

While the empirical performance of RADAR is promising, I have concerns regarding its technical novelty and the true attribution of improvement to claimed components.

---

> ### Author Response · Authors · 2025-11-26
> **Response (1/4)**
>
> >  **W1: The paper's contribution lies in the clever application of existing techniques rather than the development of fundamentally new methods. Both SVD and Sinkhorn normalization are standard, well-established algorithms. While the engineering is solid, the work feels more incremental than transformative from a methodological standpoint.**
>
> We appreciate the reviewer's perspective. While we agree that SVD and Sinkhorn normalization are techniques widely used in different domains, we clarify that RADAR’s novelty lies not in the invention of these operators, but in the principled analysis and architectural integration required to solve the neglected problem of Asymmetric VRP (AVRP). Much like impactful works in our field (e.g., MatNet [1] or LEHD [2]), which leverage standard components to unlock new capabilities, RADAR recombines known tools to address a fundamental gap in neural combinatorial optimization.
>
> Our goal with RADAR is similar in spirit for neural VRP solvers. Concretely, to the best of our knowledge, this is the first work in neural VRP that (i) explicitly frames asymmetry as a representation problem and studies how different encodings affect a model’s ability to solve asymmetric instances, (ii) systematically categorizes existing initializations into uninformed and informed schemes, (iii) decomposes asymmetric costs into static and dynamic components and, based on this analysis, motivates an SVD-based embedding for static asymmetry together with a Sinkhorn-based attention normalization for dynamic asymmetry, and (iv) broadens the discussion of asymmetry in neural routing by studying ACVRP at scale and 16 distinct asymmetric settings.
>
> Beyond the choice of SVD and Sinkhorn themselves, our contribution is primarily about what representation is needed for asymmetric VRPs and why. SVD and Sinkhorn are used as principled tools that follow directly from our static–dynamic asymmetry analysis, rather than as ad-hoc algorithmic add-ons. In addition, our study provides several new insights specific to asymmetric settings, including the role and limitations of coordinate under asymmetry and a systematic comparison between uninformed and informed initializations. Taken together, these analyses and experiments reveal that existing “euclidean-centric” practices are insufficient, and that carefully designed representations for distance matrix can substantially change model performance. We believe that while the building blocks are classical, the way they are instantiated, analyzed, and validated in the asymmetric VRP context provides new, actionable insight for the community.
>
> [1] Matrix Encoding Networks for Neural Combinatorial Optimization
>
> [2] Neural combinatorial optimization with heavy decoder: Toward large scale generalization

---

> > ### Author Response · Authors · 2025-11-26
> > **Response (4/4)**
> >
> > >  **Q1: To verify the necessity of the proposed SVD initialization, could you provide results for two models that keep all other settings unchanged (including z-score normalization) and combine Sinkhorn normalization with forward-only distance features (i.e., top-10 nearest outgoing neighbors) and bidirectional distance features (i.e., concatenating top-10 outgoing and top-10 incoming neighbor distances), respectively?**
> >
> > We thank the reviewer for this helpful suggestion, which allows us to more directly assess the contribution of the SVD-based initialization. Following the request, we implemented two additional variants that keep all settings identical to RADAR (including z-score normalization and Sinkhorn-based attention) but differ in how they encode asymmetric distances: NN-Forward, which uses only the top-10 outgoing neighbors, and NN-Bidirectional, which concatenates the top-10 outgoing and top-10 incoming neighbors. The results on ATSP are shown below. Both NN-Forward and NN-Bidirectional remain clearly worse than RADAR, especially on large instances. This confirms that simply adding local forward or bidirectional distance features with Sinkhorn is not sufficient, and that the SVD-based initialization (by compactly encoding global static asymmetry from the full distance matrix) provides a substantial additional benefit.
> >
> > |Method|100|GAP|Time|200|GAP|Time|500|GAP|Time|1000|GAP|Time|
> > |:-|:-:|:-:|:-:|:-:|:-:|:-:|:-:|:-:|:-:|:-:|:-:|:-:|
> > |NN-Forward|1.5797|0.98%|0.03m|1.5970|1.58%|0.13m|1.6491|4.62%|1.35m|1.7902|13.74%|11.06m|
> > |NN-Bidirectional|1.5790|0.94%|0.03m|1.5929|1.32%|0.13m|1.6317|3.51%|1.35m|1.7594|11.79%|11.06m|
> > |RADAR|**1.5756**|**0.72%**|0.04m|**1.5879**|**1.01%**|0.15m|**1.6098**|**2.13%**|1.45m|**1.6389**|**4.13%**|11.57m|
> >
> > >  **Q2: What is the number of iterations T used for Sinkhorn normalization?**
> >
> > Answer: We appreciate the reviewer highlighting this missing detail. In our main experiments, we fix the number of Sinkhorn iterations to T = 10.

---

> ### Author Response · Authors · 2025-11-26
> **Response (2/4)**
>
> >  **W2: In Table 1, the authors state that all retrained baselines were evaluated using z-score normalization. However, the authors do not justify why this specific normalization was uniformly applied. Applying a non-native normalization may unfairly penalize baseline performance, making it unclear if the comparison was conducted under optimal conditions for all methods.**
>
> Thank you for the constructive suggestions! Our initial adoption of z-score normalization was motivated by its standard role in deep learning: it mitigates scale sensitivity and stabilizes gradients, which is particularly critical for attention mechanisms processing raw distance matrices with varying magnitudes.
>
> To address the concern that enforcing a uniform z-score normalization across all baselines might bias the comparison, we re-ran all constructive neural solvers on the ATSP benchmarks under three strictly controlled input settings (z-score, min-max, and no normalization). Across all backbones and problem sizes, z-score was consistently competitive and most often improved performance relative to min-max and no normalization. This indicates that our use of z-score does not unfairly penalize the baselines; instead, it provides a robust common preprocessing that benefits them as well. We add these results as an ablation in the revised version and clarify that z-score was chosen as a shared, empirically validated normalization to enable a fair comparison rather than as a choice tuned specifically for RADAR.
>
> |Method|100|GAP|Time|200|GAP|Time|500|GAP|Time|1000|GAP|Time|
> |-|-|-|--|-|-|-|-|-|-|-|-|-|
> |RADAR-|1.5840|1.26%|0.04m|1.6939|7.75%|0.15m|3.7412|137.34%|1.45m|6.0139|281.42%|11.57m|
> |RADAR+|1.5776|0.85%|0.04m|1.5917|1.25%|0.15m|1.6247|3.07%|1.45m|1.6756|6.27%|11.57m|
> |RADAR*|1.5756|0.72%|0.04m|1.5879|1.01%|0.15m|1.6098|2.13%|1.45m|1.6389|4.13%|11.57m|
> |ReLD-|diverge|-|-|diverge|-|-|diverge|-|-|diverge|-|-|
> |ReLD+|1.5976|2.13%|0.03m|1.6403|4.34%|0.15m|1.8207|15.50%|1.49m|2.4648|56.32%|12.18m|
> |ReLD*|1.5900|1.64%|0.03m|1.6310|3.75%|0.15m|1.7873|13.39%|1.49m|2.0723|31.67%|12.18m|
> |Matnet_random-|diverge|-|-|diverge|-|-|diverge|-|-|diverge|-|-|
> |Matnet_random+|1.6245|3.85%|0.02m|1.6907|7.54%|0.13m|1.9651|24.67%|1.34m|2.8334|79.70%|11.11m|
> |Matnet_random*|1.5969|2.08%|0.02m|1.6543|5.23%|0.13m|1.8610|18.10%|1.34m|2.1821|38.64%|11.11m|
> |ICAM-|1.6440|5.09%|0.01m|1.9715|25.41%|0.06m|2.6485|68.02%|0.73m|3.3871|114.82%|9.80m|
> |ICAM+|1.6638|6.36%|0.01m|1.9735|25.53%|0.06m|3.0408|92.91%|0.73m|4.3716|177.26%|9.80m|
> |ICAM*|1.6580|5.99%|0.01m|1.8471|17.49%|0.06m|2.4592|56.01%|0.73m|2.9069|84.69%|9.80m|
> |One_hot-|diverge|-|-|diverge|-|-|diverge|-|-|diverge|-|-|
> |One_hot+|1.6038|2.53%|0.02m|1.6601|5.60%|0.12m|-|-|-|-|-|-|
> |One_hot*|1.5995|2.25%|0.02m|1.6727|6.40%|0.12m|-|-|-|-|-|-|
> |Matnet-|1.6473|5.31%|0.03m|3.1925|103.07%|0.14m|-|-|-|-|-|-|
> |Matnet+|1.6158|3.29%|0.03m|3.1493|100.32%|0.14m|-|-|-|-|-|-|
> |Matnet*|1.6158|3.29%|0.03m|1.9111|21.56%|0.14m|-|-|-|-|-|-|
> |ELG-|1.6080|2.79%|0.06m|1.8492|17.63%|0.27m|2.3966|52.04%|2.60m|2.5286|60.37%|22.31m|
> |ELG+|1.5907|1.69%|0.06m|1.6287|3.60%|0.27m|1.7404|10.41%|2.60m|1.9380|22.91%|22.31m|
> |ELG*|1.5982|2.17%|0.06m|1.6423|4.47%|0.27m|1.7456|10.74%|2.60m|1.8441|17.17%|22.31m|
>
>
> Note: The suffix “-” denotes no normalization, “+” denotes min–max normalization, and “*” denotes z-score normalization.
>
> >  **W3: The paper uses different normalizations for different datasets. However, the paper provides no ablation study to compare the effect of z-score versus min-max (or no normalization) on the same dataset. It remains unclear whether the impressive performance gains are fully attributable to the proposed SVD and Sinkhorn modules or are partially an artifact of a specific, un-analyzed normalization choice.**
>
>
> We thank the reviewer for raising this point about normalization. In our experiments, normalization is always aligned within each table: all methods in a given table use the same normalization on a given dataset. Except for Table 1, where we deliberately apply z-score norm to all constructive neural solvers, other tables strictly follow the normalization used in the original paper for that dataset as per the original authors' suggestion, so RADAR is never compared against baselines that use a “weaker” normalization in the same setting. This design is precisely to ensure that our gains cannot be attributed to mixing different norms within a comparison.

---

> ### Author Response · Authors · 2025-11-26
> **Response (3/4)**
>
> >  **W4: The ablation study in Table 6 severely weakens the case for the SVD-based initialization as a standalone contribution. The RADAR-SVD performs very poorly on larger-scale problems, achieving a gap of over 20% on ATSP1000. This poor generalization suggests that the strong performance is critically dependent on the Sinkhorn operation.**
>
> Thank you for the observation! The earlier table only showed the “SVD-only” variant at large scale, which indeed gives the impression that SVD is not useful as a standalone component. In the rebuttal we therefore provide a complete 2×2 ablation (SVD on/off × Sinkhorn on/off).
>
> |Method|SVD|Sink|100|Gap|Time|200|Gap|Time|500|Gap|Time|1000|Gap|Time|
> |-|-|-|-|-|-|-|-|-|-|-|-|-|-|-|
> |RADAR|✗|✗|1.5969|2.08|0.02m|1.6543|5.23|0.13m|1.8610|18.06|1.34m|2.1821|38.64|11.11m|
> |RADAR|✓|✗|1.5928|1.82|0.03m|1.6273|3.51|0.14m|1.7418|10.50|1.44m|1.9342|22.89|11.45m|
> |RADAR|✗|✓|1.5829|1.19|0.03m|1.6007|1.82|0.13m|1.6379|3.91|1.43m|1.6878|7.24|11.37m|
> |RADAR|✓|✓|**1.5756**|**0.72**|0.04m|**1.5879**|**1.01**|0.15m|**1.6098**|**2.13**|1.45m|**1.6389**|**4.13**|11.57m|
>
> From this table, we observe that (i) adding SVD alone already improves over the plain RADAR backbone without SVD or Sinkhorn across all sizes, and (ii) SVD further improves performance when combined with Sinkhorn. At the same time, we agree with the reviewer that large-scale generalization is primarily driven by Sinkhorn; we do not expect an initialization scheme by itself to fully counter distribution shift. Instead, the goal of the SVD-based initialization is to mitigate the information loss of existing “informed” initializations by compressing asymmetric distance information via eigen-structure, so that the model starts from a representation that preserves more of the static asymmetry. The new ablation confirms this complementary role: Sinkhorn is crucial for robustness, while SVD provides consistent, non-trivial gains both with and without Sinkhorn. We will revise the paper to include this complete ablation and to more clearly position SVD as a principled, supporting component rather than the sole source of RADAR’s generalization.
>
> >  **W5: Some critical hyperparameters' details are missing. For example, the number of iterations T for Sinkhorn normalization is not specified. It is unclear whether the increasing number of iterations significantly increases the computational load and performance of the model.**
>
> We thank the reviewer for pointing out the missing hyperparameter specification. In all main experiments we set the number of Sinkhorn iterations to T = 10. For the rebuttal, we additionally ran a sensitivity study on ATSP with T = 1, 5, and 10. The results show that using more iterations steadily improves solution quality, while the corresponding increase in runtime is marginal. Thus, higher T brings clear accuracy gains at almost no extra computational cost. We will report these results in the revised version and clearly state that T = 10 is our default choice.
>
> |Method|Iter|100|Gap|Time|200|Gap|Time|500|Gap|Time|1000|Gap|Time|
> |-|-|-|-|-|-|-|-|-|-|-|-|-|-|
> |RADAR|1|1.5815|1.10%|0.04m|1.6005|1.81%|0.14m|1.6542|4.94%|1.43m|1.7576|11.67%|11.42m|
> |RADAR|5|1.5764|0.77%|0.04m|1.5909|1.20%|0.14m|1.6210|2.84%|1.44m|1.6590|5.41%|11.50m|
> |RADAR|10|**1.5756**|**0.72%**|0.04m|**1.5879**|**1.01%**|0.15m|**1.6098**|**2.13%**|1.45m|**1.6389**|**4.13%**|11.57m|

---

### Meta-Review · Area_Chair_443N · 2025-12-23

**Summary:**

The reviewers appreciated the combination of two techniques to address "static asymmetry" and "dynamic asymmetry". They raised a variety of concerns questioning the novelty of the techniques, the empirical evaluation (e.g., normalization choice, missing details about hyperparameter values, relevancy of chosen baselines, comparison with OR heuristics, source of performance gain on larger instance, ablation study), the extension to other combinatorial optimization problems, or the dependence on attention-based architecture.

**Reviewer Concerns:**

I think that the concerns raised by the reviewers have been addressed thoroughly by the authors in their rebuttal. I appreciate the depth of their answers where they corrected some misunderstandings, provided numerous additional experimental results to support their claims, or suggest methodologies to tackle less related questions.

**Reviewer Scores:**

Given the quality of the rebuttal, I believe that all the scores would have been at least 6.

---

### Decision · Program_Chairs · 2026-01-26

Accept (Poster)